

# Effect of trachyte and basalt rock powders on maize (Zea mays L.) growth and yield on Fluvisols in Cameroon's Sudano-Sahelian zone (Central Africa)

Bienvenu Sidsi[1,3], Claudine Vounba[1], Simon Djakba Basga[2], Aubin Nzeugang Nzeukou[3], Merlin Gountie Dedzo[4], Désiré Tsozué[3*]

[1] Institute of Agricultural Research for Development (IRAD), P.O. Box 33 Maroua, Cameroon
[2] Institute of Agricultural Research for Development (IRAD), P.O. Box 2123 Yaoundé, Cameroon
[3] Department of Earth Sciences, Faculty of Science, University of Maroua, P.O. Box 814 Maroua, Cameroon
[4] Department of Life and Earth Sciences, Higher Teachers' Training College, University of Maroua, P.O. Box. 55, Maroua, Cameroon

*Correspondence to*: Désiré Tsozué (tsozudsir@yahoo.fr)

**Abstract.** The Sudano-Sahelian zone of Cameroon, characterized by a low annual rainfall, faces challenges in soil fertility preservation due to agricultural intensification and unsustainable practices. This study aims to evaluate the effect of trachyte and basalt powders inputs on soil and maize yield in Guiring experimental farm. Fieldwork involved collecting and describing samples of trachyte, basalt, and soil and setting up the experimental design. In the laboratory, the ground rock samples underwent geochemical analysis, and the soil samples were analysed for their mineralogical and physicochemical properties. The experiment followed a completely randomized block design with six treatments ($T_0$, $T_1$, $T_2$, $T_3$, $T_4$, and $T_5$) and four replications. Growth and yield parameters of maize, include germination rate, plant height, number of leaves per plant, stem diameter, ear length, ear diameter, ear weight, 100-grain weight, and grain yield (kg ha$^{-1}$). The soil consists of kaolinite, smectite, sepiolite, and quartz. Its texture is dominated by sand fraction, with a neutral pH (7.0). The organic matter (2.6±0.67%) and total nitrogen contents (0.1±0.0%) are relatively low. The concentrations of potassium, magnesium, sodium, and calcium are 0.2±0.1 cmol$_c$ kg$^{-1}$, 2.5±1.6 cmol$_c$ kg$^{-1}$, 0.3±0.2 cmol$_c$ kg$^{-1}$, and 3.9±1.5 cmol$_c$ kg$^{-1}$, respectively. The cation exchange capacity is moderate to high (22.1±2.5 cmol$_c$ kg$^{-1}$), while the available phosphorus content is high (19±7.0 mg kg$^{-1}$). This soil is classified as Ochric Dystric Fluvisols according to the WRB. These soil characteristics are moderately suitable for maize cultivation. Fertilization trials showed a significant improvement in maize growth and yield, within plots treated with basalt powder yielding higher (2558.6 kg ha$^{-1}$ and 2931.2 kg ha$^{-1}$) than those treated with trachyte powder (2362.9 kg ha$^{-1}$ and 2763.9 kg ha$^{-1}$) and the control plots (645.8 kg ha$^{-1}$). Plots treated with NPK fertilizer recorded the highest yield (3164.5 kg ha$^{-1}$). Although the treatment with conventional fertiliser resulted in a relative higher yield, the advantage of using rock powders lies in their environmental benefits, long-term effectiveness, and more affordable cost.



## 1 Introduction

In recent decades, population growth and food insecurity have led to intensification of agriculture, soil overexploitation, and excessive use of synthetic fertilizers (Pham et al., 2018; Jhariya et al., 2021; Khatri et al., 2024). Intensively exploited soils degrade and impoverish rapidly, resulting in a reduction of biological activity, soil acidification and deterioration of soil structure (Kuria et al., 2018; Nanganoa et al., 2019; Rajwar et al., 2021). As a limited resource, it takes between 200 to 1000 years to form a layer of 2.5 cm in thickness, making its exploitation for agricultural purposes fragile due to the increase in the

world population and climate change (Moges and Taye, 2016). Soil degradation has become a major global concern due to its harmful effects on biodiversity, agricultural production, and food security (Mekuriaw et al., 2017; Hossain et al., 2020). To maintain soil fertility and productivity, it is necessary to replenish the supply of nutrients (Nkouathio et al., 2008; Garai et al., 2022).

Soil provides physical stability, support for agriculture and contributes to the nutrient cycle, water retention capacity, storage,

filtering, buffering, and transformation of compounds (Tsozué et al., 2020a; Toor et al., 2021; Sivaram et al., 2023). Soil health, also known as soil quality, is a key factor in sustainable agriculture, influencing ecosystem quality, such as air and water quality (Are et al., 2018; Tsozué et al., 2020a). Maintaining soil quality is an essential determinant for managing ecosystem services and sustainable soil management (Tsozué et al., 2020a; Wu et al., 2023). Its restoration and assessment require the use of relevant biological, physical, and/or chemical indicators (Tsozué et al., 2020a; Oliveira et al., 2024). It is important to

consider current land use and precisely define the type of disturbance that may alter its quality (Peng et al., 2021). In the case of anthropogenic disturbances, land use, including agricultural management systems, strongly influences soil quality, expressed by changes in its physicochemical and biological properties and agricultural yield (Farahani et al., 2019; Tsozué et al., 2020a; Mamabolo et al., 2024).

Intensive agriculture leads to excessive nutrient depletion from the soil, thereby compromising its fertility on a global scale

(Padhiary et al., 2024). While nitrogen (N) and phosphorus (P) are often highlighted, potassium (K) also plays a crucial role, and its depletion is becoming an increasing concern (Khan et al., 2023). To address this issue, the use of rock powders derived from base-rich silicate rocks has emerged as a sustainable alternative to synthetic fertilizers. These natural fertilizers enrich the soil with essential nutrients and enhance its structure, although their performance depends on soil characteristics and climatic conditions (Manning and Theodoro, 2020; Rodrigues et al., 2024b). Numerous studies have explored the use of silicate

rock powders for soil fertilization and remineralization. In the Americas, several studies have demonstrated that these mineral powders enhance water retention and soil fertility while contributing to carbon sequestration (Beerling et al., 2018; Theodoro et al., 2021; Lewis et al., 2021; Ramos et al., 2022; Medeiros et al., 2023). In Europe, research has confirmed their positive impact on improving soil fertility and remineralization (Beerling et al., 2018; Vienne et al., 2022). Similarly, in Asia, silicate rocks have shown their ability to increase agricultural productivity (Xiang et al., 2020; Basak et al., 2023; Sniatala et al., 2023).

Africa has also embraced this innovation, using rock powders to boost agricultural yields while preserving the environment (Mekuriaw et al., 2017; Muhie, 2022). In Cameroon, in particular, several studies have demonstrated the effectiveness of these



powders (Nkouathio et al., 2008; Mein'da et al., 2022; Tetchou et al., 2022). Although crushed rocks offer agronomic and environmental benefits, some limitations remain, including their slow dissolution rate, high logistical costs, and the need for pedological expertise to maximize their effectiveness. Given the growing concerns over soil degradation and nutrient depletion,

it becomes crucial to explore sustainable solutions, particularly in regions like the Sudano-Sahelian zone of Cameroon, where challenging climatic conditions contribute to the deterioration of soil fertility.

The Sudano-Sahelian zone of Cameroon, characterized by a low precipitation and a prolonged dry season, faces challenges in preserving soil fertility. Inadequate agricultural practices and pressure on the soil have led to its degradation, resulting in a decline in fertility and agricultural yield (Prasannakumar et al., 2011; Hishe et al., 2017). According to FAO (2013), global

food demand is expected to increase by 60% between 2006 and 2050, making imperative, if not essential, to modify or replace agricultural techniques with more sustainable and environmentally friendly practices to preserve the integrity of various environmental components and ensure long-term agricultural sustainability (Nkouathio et al., 2008; Muhie, 2022). One of these techniques involves effectively regenerating soil fertility using appropriate methods that do not impact the environment, including the use of geological materials (Burbano et al., 2022). As the minerals constituting rocks dissolve, the released

chemical elements become available to plants (Silva et al., 2005; Manning and Theodoro, 2020; Basak et al., 2023). To address the challenges of soil fertility decline and rising fertilizer costs in the Sudano-Sahelian zone, exploring alternative solutions such as local rock powders becomes essential for sustainable and resilient agriculture.

The rising cost of fertilizers represents a major constraint for farmers, especially in the Sudano-Sahelian zone of Cameroon, limiting their access to these essential inputs (Sinha et al., 2022). In this region, synthetic NPK fertilizer is generally used in

combination with urea, which has become a standard treatment for most crops. For maize production, this standard fertilisation is the guideline recommended by the Center for International Forestry Research and World Agroforestry (CIFOR-ICRAF) in the framework of the Strengthening Innovation Systems in the North of Cameroon (ReSI-NoC) projet 2020-2024 (CIFOR-ICRAF, 2024). This choice is driven both by the severe soil degradation in the area and by local farming practices, which are characterized by the application of relatively high doses to compensate for the low fertility of the land. This situation has raised

global concerns about food shortages, forcing countries reliant on fertilizer imports to find mechanisms and new technological pathways to reduce their dependence on the international market (Xiang et al., 2020; Sniatala et al., 2023). The use of local silicate rock powders, or soil remineralizers, appears to be a promising alternative (Dalmora et al., 2020; Swoboda et al., 2023). Trachyte and basalt powders have been specifically selected for their complementary geochemical properties and local availability within the study area. Basalt, a mafic rock, is rich in Ca, Mg, Fe, and trace elements, while trachyte, which is more

siliceous, has a relatively higher content of K (Manning, 2010; Anda et al., 2015). These characteristics make them natural amendments capable of gradually releasing essential nutrients for plant growth, while improving the soil's physico-chemical properties over the long term. This agroecological approach, based on the use of local geological resources, not only reduces reliance on synthetic fertilizers but also sustainably improves soil fertility while being more economical and environmentally friendly (Ramos et al., 2017; Conceição et al., 2020; Swoboda et al., 2022). In light of the challenges posed by increasing





fertilizer costs and the need for sustainable agricultural practices, exploring the potential of trachyte and basalt powders to enhance soil fertility and maize production in Cameroon has become a vital research focus.

Maize is one of the most produced and consumed cereals in the world, with approximately 1016 million tons cultivated across nearly 184 million hectares (Eshetie, 2017). It is primarily used for food, but also plays a key role in the production of biofuels, as well as the extraction of starch, oil, and other industrial substances (Ranum et al., 2014). In the northern regions of Cameroon in particular, maize is a staple food in nearly every household (Chimi et al., 2023). However, the national average maize yield in Cameroon remains relatively low, ranging from 1.7 to 2.5 tons per hectare well below its agronomic potential, which is estimated at 5 to 6 tons per hectare under optimal conditions (Tsafack et al., 2024). This yield gap is largely attributed to poor soil fertility, limited access to agricultural inputs, and inadequate soil management practices. In a context marked by ongoing soil degradation, the environmental impacts of intensive chemical fertilizer use, and rising input costs, it is crucial to explore the extent to which locally available geological materials such as trachyte and basalt powders can improve soil fertility, enhance maize growth and yield, and support a more sustainable and climate-resilient form of agriculture. This issue lies at the core of the research conducted at the experimental farm of the Agricultural Research Institute for Development (IRAD) in Guiring.

## 2 Materials and methods

### 2.1 Study area

This study was conducted in the locality of Guiring, situated in the Diamaré Division (Maroua III Subdivision) in the Far-North Region of Cameroon, located between 10°32′21.8″ and 10°41′55.7″ North latitude and 14°15′33.8″ and 14°29′42.0″ East longitude (Fig. 1.). The area has a Sudano-Sahelian climate (Suchel, 1987), with a short and irregularly distributed rainy season of four months from June to September (Aridity index > 20), followed by a long dry season of eight months lasting from October to May (Aridity index < 20). The average annual rainfall is approximately 792.25 mm, and the average annual temperature reaches 27.7°C (Table 1). (Table 1). The highest temperatures are recorded in April, while the lowest occur in January. The terrain is mainly characterized by a relatively flat topography, marked by several ecological zones, including the Sahelo-Sudanian domain, the elevated Sudanian domain, and the periodically flooded Sahelian domain. The substratum consists of alluvial deposits of Quaternary age (Tament et al., 2015; Gountié et al., 2019). The vegetation consists of scattered trees and shrubs associated with grasses. Among the most representative species are *Acacia seyal*, *Balanites aegyptiaca*, *Faidherbia albida*, and *Ziziphus mauritiana* (Letouzey, 1980).



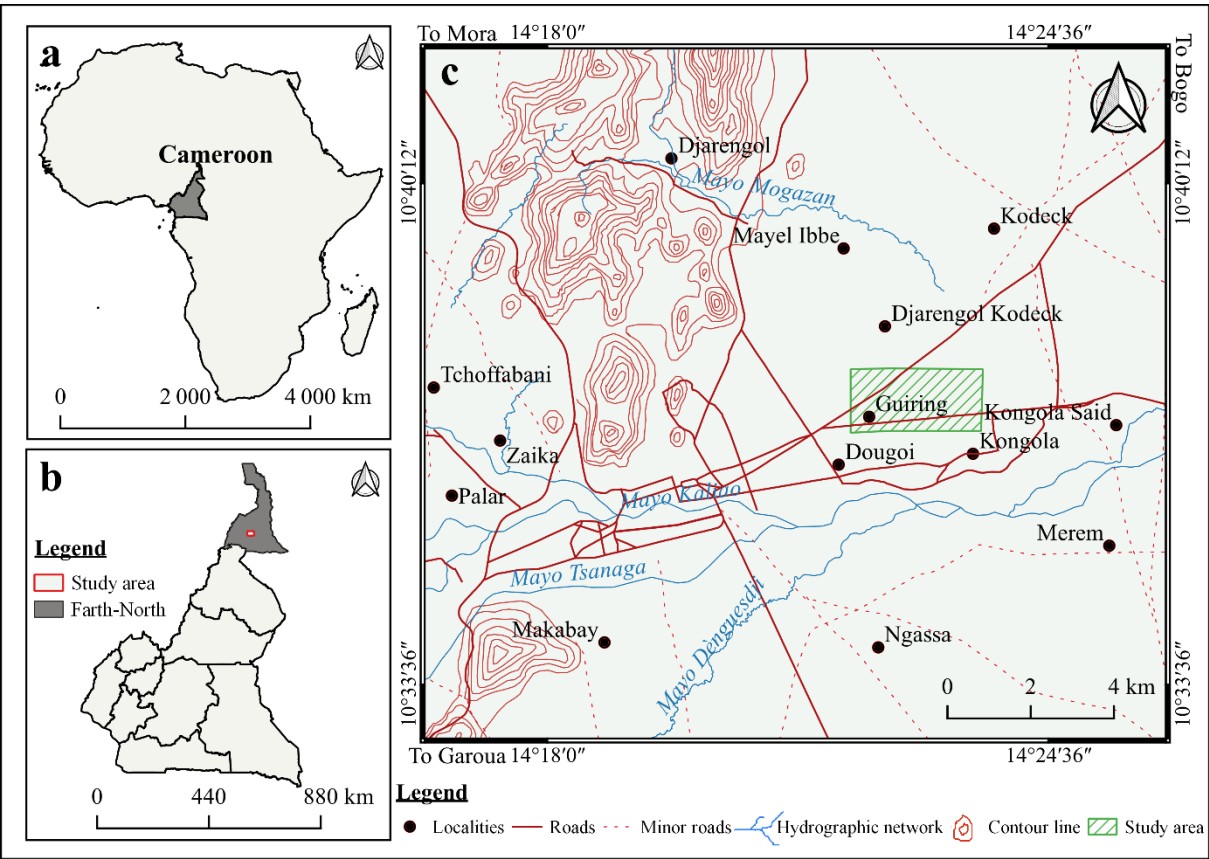

**Fig. 1. (a)** Location of Cameroon in Africa, **(b)** Location of Far-North region of Cameroon, **(c)** Location of the study area.

*Table 1. Monthly averages of precipitation, temperature, and relative humidity in Maroua-Salack from 1980 to 2020.*

| Month | Jan | Feb | Mar | Apr | May | Jun | Jul | Aug | Sep | Oct | Nov | Dec | Mean | Total |
|---|---|---|---|---|---|---|---|---|---|---|---|---|---|---|
| Precipitation (mm) | 0.0 | 0.0 | 0.9 | 32.1 | 60.5 | 105.8 | 192.8 | 232.4 | 118.2 | 48.8 | 0.5 | 0.3 | / | 792.3 |
| Temperature (°C) | 24.2 | 27.5 | 30.9 | 32.9 | 31.5 | 28.9 | 26.5 | 25.2 | 25.9 | 27.5 | 27.0 | 24.4 | 27.7 | / |
| Relative humidity (%) | 19.0 | 14.0 | 11.0 | 20.0 | 40.0 | 60.0 | 73.0 | 81.0 | 79.0 | 56.0 | 27.0 | 23.0 | 42.0 | / |
| Aridity index | 0.0 | 0.0 | 0.3 | 9.0 | 17.5 | 31.8 | 63.4 | 79.2 | 30.9 | 15.6 | 0.2 | 0.1 | 20,6 | / |

**2.2 Experimental design, treatments, and plant material**

The experimental design is a completely randomized block design with six (06) treatments ($T_0$, $T_1$, $T_2$, $T_3$, $T_4$, $T_5$) as described in table 2 and four replicates, resulting in twenty-four plots (6 x 4) (Fig. 2.). The choice of a completely randomized block design accounts for environmental heterogeneity and variability in the soil's natural fertility by randomly distributing treatments within each block, thereby minimizing the influence of uncontrolled factors. It was conducted at the experimental

site of IRAD in Guiring. Each experimental unit received one of the recommended treatments, $T_0$, $T_1$, $T_2$, $T_3$, $T_4$, and $T_5$ (Table 2). Treatment $T_0$ corresponds to the plot that received no input. It serves as the initial reference for evaluating the effects of





the various treatments. To enable an objective and effective comparison of the treatments, a standard treatment based on synthetic NPK fertilizer combined with urea ($T_5$) was included, also representing a control, a reference fertilization practice in the region recommended by CIFOR-ICRAF (2024). The maize density was 0.8 x 0.3 m, with 2 to 3 seeds needed per hole.

Two weeks after sowing, thinning was performed to leave one plant per cluster. In each experimental unit, a yield square was demarcated by eliminating plants located at the edges. Parameters were measured on the three central lines, and the four best plants from each line were selected, for a total of 12 plants per plot. A total of 288 plants were retained for all replicates. The plant material used is the CMS-9015 variety of maize, developed by IRAD, which is an open-pollinated variety. This variety is specifically adapted to the Sudanese-Sahelian zone of Cameroon. Under normal conditions, it produces grain yields ranging

from 4 to 4.5 tons per hectare, with a vegetative cycle of 95 days. Manual weeding was carried out throughout the experiment, based on the appearance and development of weeds, to prevent excessive competition with the crops.

*Table 2. Summary and nature of the experimental treatments.*

| Treatment designation | Description of the treatment |
|---|---|
| $T_0$ | Control plot without fertilizing inputs |
| $T_1$ | Treatment with trachyte powder applied directly after sowing (2000 kg ha$^{-1}$; 2.0 kg/9.6m$^2$) |
| $T_2$ | Treatment with trachyte powder applied directly after sowing (2000 kg ha$^{-1}$; 2.0 kg/9.6m$^2$ ) + urea (104.2 kg ha$^{-1}$; 100 g/9.6m$^2$) at 06 weeks |
| $T_3$ | Treatment with basalt powder applied directly after sowing (2000 kg ha$^{-1}$; 2.0 kg/9.6m$^2$) |
| $T_4$ | Treatment with basalt powder applied directly after sowing (2000 kg ha$^{-1}$; 2.0 kg/9.6m$^2$) + urea (104.2 kg ha$^{-1}$; 100 g/9.6m$^2$) at 06 weeks |
| $T_5$ | Treatment with NPK fertilizer applied two weeks after sowing (208,3 kg ha$^{-1}$; 200 g/9.6m$^2$) + urea (104.2 kg ha$^{-1}$; 100 g/9.6m$^2$) at 06 weeks (Reference fertilization practice in the region) |



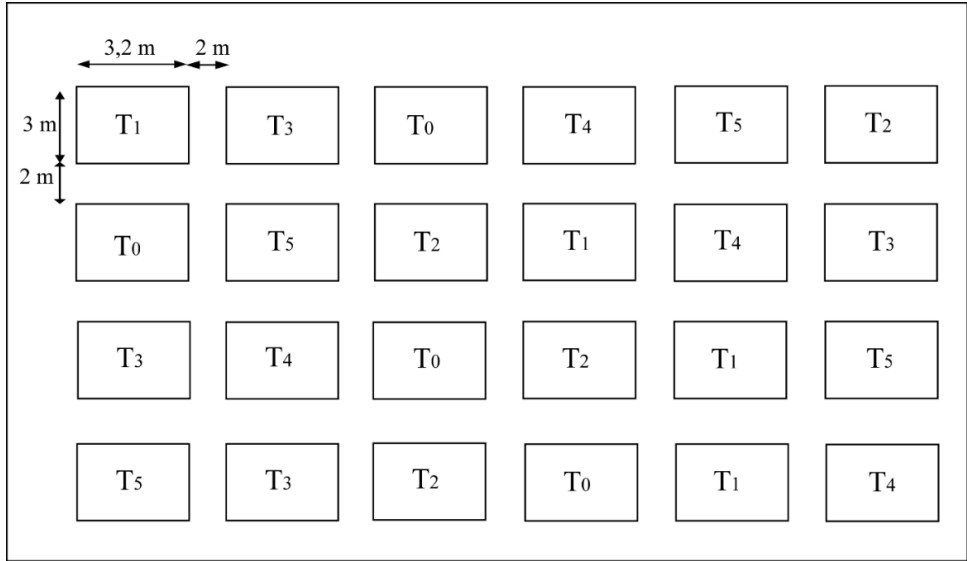

**Fig. 2.** Completely randomized block design with four replications.

## 2.3 Rock sampling and analysis

Samples of trachyte (Fig. 3 a) were collected from the localities of Zamai, at 10°38′06″ north latitude and 13°53′55″ east longitude and that of basalts (Fig. 3 b) from the locality of Gawar, at 10°30′50.7″ north latitude and 13°50′25.68″ east longitude.
Approximately 500 kg of each type of rock were sampled. These samples were then carefully transported for grinding, thin section preparation, petrographic analyses, and geochemical analyses. In total, 3 samples of trachyte and 3 samples of basalts were subjected to petrographic and geochemical analysis in the laboratory.

The rock sample was fragmented and passed through a ball mill, then sieved using a 2 mm mesh to obtain powder less than 2 mm (Figs. 3 c and d). This powder was composed of 49%, 22% and 29% of fractions passing through 0.8 mm, 0.3 mm, and
0.1 mm sieves respectively. In total, 100 kg of trachyte powder and 100 kg of basalt powder were obtained.

The petrographic analysis was carried out on thin sections prepared at the Tokyo Institute of Technology, Japan. The microscopic description was done in the geology laboratory at the University of Maroua (Cameroon) using an optic microscope to examine the rock samples in detail. This phase allowed for the precise identification of the different minerals present in each sample, based on their optical and textural characteristics. The observations focused on mineralogical assemblages, grain
shapes and the relationships between the various minerals.

For the measurements of trace and major elements, all samples were crushed into millimetre-sized fragments using a jaw crusher, and fragments with saw and hammer marks were carefully separated. After each sample, the jaw crusher was systematically cleaned with compressed air and deionized water. The major elements and some trace elements, including Scandium (Sc), Cobalt (Co), Vanadium (V), Nickel (Ni), and Chromium (Cr), were analysed by X-ray fluorescence (XRF)
using a Rigaku RIX2100 (manufactured by Rigaku Corporation, Japan) at Osaka City University, Graduate School of Science



in Japan. The trace element concentrations in the samples were determined by ICP-MS (manufactured by Thermo Fisher Scientific, USA) at the Tokyo Institute of Technology, Japan, in accordance with the method described by Yokoyama et al. (1999). Approximately 50 mg of rock powder were digested with Nitric Acid ($HNO_3$), Perchloric Acid ($HClO_4$), Hydrochloric Acid (HCl), and Hydrofluoric Acid (HF). Before acid digestion, enriched spikes of 113In and 203Tl were added to the sample

to serve as internal standards during measurement. The data reported by Makishima and Nakamura (2006) were used to calibrate the trace element concentrations against the basalt and rhyolite standards JB-3 and JR-1, respectively. The typical analytical reproducibility (2σ) was 4% for Niobium (Nb) and Lead (Pb), 5% for Yttrium (Y) and Tantalum (Ta), and <3% for other trace elements.

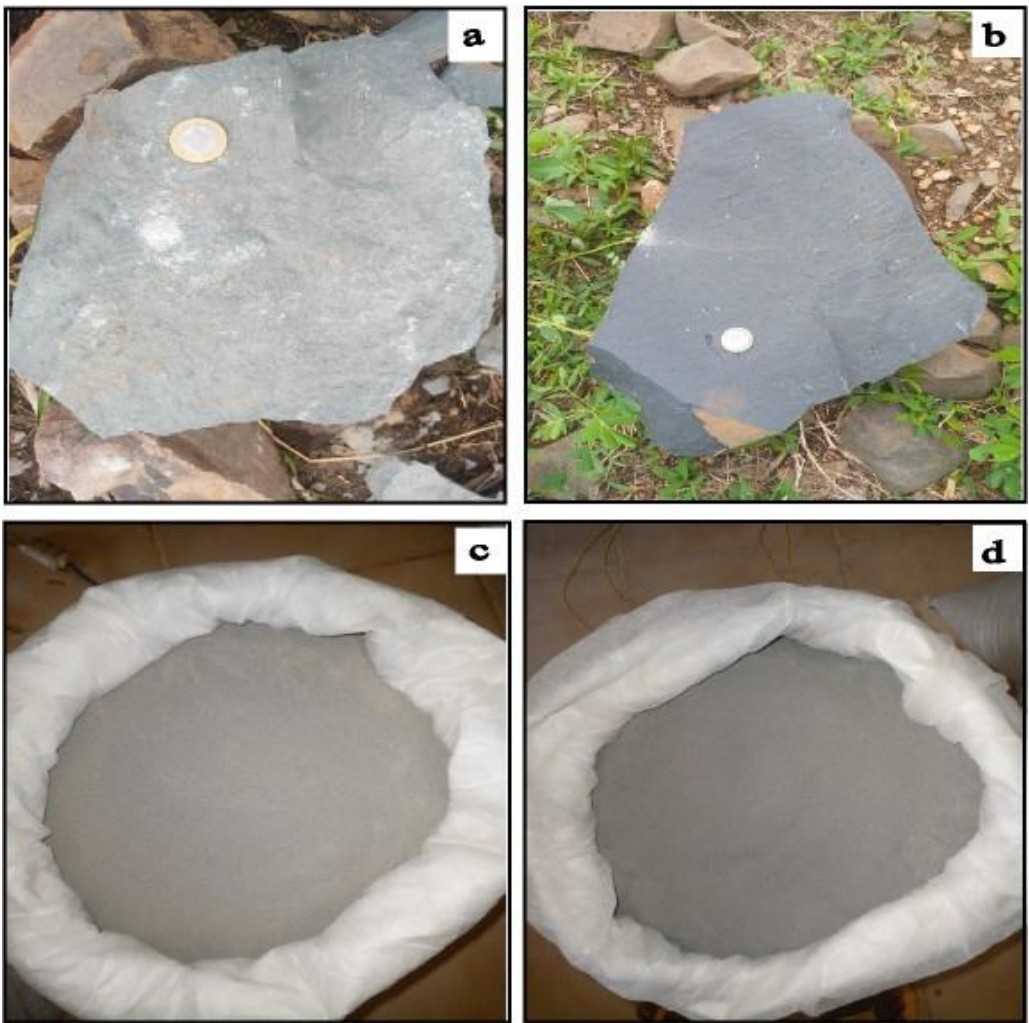

**Fig. 3**. **Samples and powders of trachyte and basalt: (a)** Trachyte sample from Zamai, **(b)** Basalt sample from Gawar**, (c)** Trachyte powder obtained after grinding, **(d)** Basalt powder obtained after grinding.



## 2.4 Soil sampling and laboratory analysis

The fieldwork was carried out in two phases. The first phase consisted of direct observation and a detailed description of the study site and its surroundings. In the second phase, a representative soil profile (10°33'17" north latitude, 14°22'16" east longitude, 386 m of altitude) with a thickness of 270 cm was excavated in a flat relief, under highly anthropogenized vegetation, and described in situ following the guidelines for soil profile description according to Baize and Jabiol (2011). The soil description took into account several parameters: colour, porosity, texture, coarse elements, structure, biological activity, thickness, and the boundaries between horizons. The colour was determined in its dry state using the Munsell Soil Colour Charts (2009). A total of six (06) samples, each weighing approximately 0.5 kg, were collected from the different horizons of the profile. The collected samples were placed in polyethylene bags, carefully labelled, and prepared for laboratory analysis.

In the laboratory, the main analyses focused on mineralogical and physicochemical analyses. Particles size distribution was performed using the Robinson pipette method on air-dried, ground, and 2 mm-sieved samples. The pH was measured in a soil-water suspension, then in a normal potassium chloride solution at 1:2.5 ratio using a glass electrode pH meter. Exchangeable bases (calcium: $Ca^{2+}$, magnesium: $Mg^{2+}$, potassium: $K^+$, and sodium: $Na^+$) were extracted from the soil using a pH 7 ammonium acetate solution and measured by atomic absorption spectrophotometry. The total exchangeable bases (TEB) was determined by adding the concentrations of $Ca^{2+}$, $Mg^{2+}$, $K^+$, and $Na^+$. Cation exchange capacity (CEC) was measured using an ammonium acetate solution at pH 7, following a three-step process: washing the soil with alcohol to remove the ammonium solution saturating the pores, determining $NH_4^+$ by Kjeldahl distillation after quantitative desorption with KCl. The CEC of the clay fraction was calculated from the CEC at pH 7, organic carbon content, and clay percentage. Base saturation (BS) was calculated using the formula (TBE/CEC) × 100. Total nitrogen (N) was determined by the Kjeldahl method. Organic carbon (OC) was measured by wet oxidation according to Walkley and Black (1934) using a sodium dichromate and concentrated $H_2SO_4$ mixture. Organic matter (OM) content was calculated by multiplying the OC by 1.724, and the C/N ratio was determined. Available phosphorus was measured using the Bray no. 2 method (Bray and Kurtz, 1945). Soil classification was done according to IUSS Working Group WRB (2022).

Mineralogical analysis was carried out using infrared (IR) spectroscopy, a non-destructive analytical method widely used to identify and characterize the functional groups present in samples. This technique provides detailed information about the chemical bonds and molecular structures of minerals. Diffuse reflectance IR spectra were recorded in the range of 4000 to 400 $cm^{-1}$ using a Perkin Elmer 2000 FTIR spectrometer (Perkin Elmer, Waltham, MA, USA), equipped with a DTGS (deuterated triglycine sulphate) detector, optimized for high sensitivity and precise resolution. To avoid contamination from atmospheric moisture and ensure the reliability of the results, measurements were carried out under strictly controlled environmental conditions. This method not only allows for the identification of minerals present in the soil but also helps describe their local chemical environments, providing a deeper understanding of the composition and properties of soils.

## 2.5 Determination of quantities of nitrogen, phosphorus and potassium supplied



The percentage of nitrogen, potassium and phosphorus in rock powder was calculated using the chemical composition of the rock. To calculate the quantity of each element in the NPK fertiliser, the percentages indicated on the label (14-24-14), the molar masses of these elements and the quantity of fertiliser were used. The percentage of nitrogen in urea was calculated using the amount of nitrogen and the molar mass of urea. The amounts of nitrogen, phosphorus and potassium used are shown in Table 3.

**Table 3.** *Quantities of nitrogen, phosphorus and potassium supplied*

| Treatment | N (kg ha$^{-1}$) | P (kg ha$^{-1}$) | K (kg ha$^{-1}$) |
|---|---|---|---|
| Trachyte | 0 | 0 | 95.2 |
| Basalt | 0 | 22.4 | 31.6 |
| NPK 14-24-14 | 29.2 | 21.8 | 24.2 |
| Urea | 48.6 | 0 | 0 |

**2.6 Pedoclimatic Assessment**

The climatic and pedological assessment of the study area was based on climatic data (precipitation, temperature, relative humidity, and insolation) collected at the Maroua-Salak station (Cameroon, 10°27'0" north latitude and 14°15'0" east longitude) between 1980 and 2020, in accordance with the climatic requirements of crops (Sys, 1985; Sys et al., 1993; Issiné

et al 2022). A climatic index (CI) was calculated using the parametric formula (Sys, 1985):

$$CI = R\min \times \sqrt{\frac{A}{100} \times \frac{B}{100} \times \ldots},$$

where Rmin is the smallest value, A, B, … are the other parametric values. Adjustments are made if the CI falls between 25 and 92.5.

The pedological assessment used a table of soil characteristics to determine a land index, based on similar parametric values

using the formula (Sys, 1985):

$$LI = R\min \times \sqrt{\frac{A}{100} \times \frac{B}{100} \times \ldots},$$

The calculation of the CI and LI allows for an integrated assessment of agroclimatic and soil quality for agricultural production.

**2.7 Collection of maize parameters**

The growth and yield parameters of maize, including germination rate, plant height, number of leaves per plant, stem diameter,

ear length, ear diameter, ear weight, 100-grain weight, and grain yield (kg ha$^{-1}$), were measured. To minimize the influence of edge effects on the average value of the response variables of interest, only the plants located at the centre of each experimental plot were considered for measurements. The germination rate was assessed 10 days after sowing by counting the number of germinated seeds in each plot. It was calculated using the following formula (Ellis et al., 1986):



$$Germination\ Rate\ (\%) = \frac{Number\ of\ germinated\ seeds}{Total\ number\ of\ seeds\ sown}\ x\ 100$$

Plant height was measured using a tape measure, recording the distance from the soil to the top of the stem, with measurements taken every 21 days. These measurements were performed on a representative sample from each plot an average value was considered (Cox and Andrade, 1988). The number of leaves was counted by tallying the leaves deployed on the main stem, with an average estimation made every 3 weeks (Cox and Andrade, 1988). The stem and ear diameter were measured using a calliper respectively before and after harvest (Ngoune Tandzi et al., 2019). The ear length, measured from base to tip, was

recorded with a tape measure when the ears were mature. The weight of the ears was obtained by weighing them with a high-precision electronic scale after harvest (Muthaura et al., 2017). The weight of 100 randomly selected dried maize grains was also measured using a precision electronic scale (Soleymani, 2018). The yield per hectare was estimated by harvesting a representative portion of each plot, weighing the harvest, and then extrapolating the results to the hectare scale using a conversion factor (Soleymani, 2018).

**2.8 Data Analysis**

Statistical analyses focused on the physicochemical properties of soils as well as the growth and yield parameters of maize. The physicochemical properties of soils were subjected to standard statistical analyses, including mean, maximum-minimum values, standard deviation, and coefficient of variation. The normality of the distribution was assessed using the Anderson-Darling test. Relationships among the various soil parameters were examined using Spearman's correlation test, with a

significance threshold set at $p < 0.05$. Since the data distribution did not follow a normal law, a logarithmic transformation was applied to measured plant parameter values (Daumas 1982). Subsequently, the data from plants within the same plot were averaged to produce a single value per replicate, thus representing an independent experimental unit. This approach aimed to avoid pseudo-replication in the statistical analysis. A two-factor analysis of variance (two-way ANOVA) was then performed to assess the existence of statistically significant differences ($p < 0.05$) between treatments. In case of significant differences,

a Tukey's HSD post hoc test was conducted to precisely identify the groups showing differences. All statistical analyses were carried out using the R software (R Core Team, version 4.5.0).

**3 Results**

**3.1 Petrographic and geochemical characteristics of rocks**

Petrographically, trachyte outcrops in the form of slabs in the Zamai locality and exhibits an aphyric texture, with rare phenocrysts (≤5%). These scarce phenocrysts are mainly composed of amphibole (Fig. 4a), sanidine (Fig. 4b), sometimes associated with clinopyroxene. Sanidine, characterized by elongated, euhedral prisms reaching up to 0.6 × 1.8 mm, displays



simple Carlsbad twins (Fig. 4b). The trachytes are composed of feldspar microliths, accompanied by a low proportion of clinopyroxene and oxides (Figs. 4 a and b). Basalts outcrop as lava flows, exhibiting well-defined columnar joints and a

porphyritic texture. Plagioclase, olivine, and clinopyroxene constitute the phenocrysts (Figs. 4 c and d). Olivine, present in some samples, is often altered to iddingsite. Peridotite xenoliths are also observed in some samples, containing approximately 65% olivine and 35% pyroxene. Plagioclase microliths, associated with clinopyroxene, olivine, and oxides, are the main components of the basalt matrix. Microliths and volcanic glass constitute the groundmass of basalts and trachytes.

From a geochemical perspective, the concentration of major and trace elements in the studied rocks are summarized in Table

3. Trachyte exhibit variations in major element contents across samples. The $SiO_2$ content ranges between 65.3% and 66.5%. Meanwhile, the $TiO_2$ content varies from 0.2% to 0.3%. The $Al_2O_3$ content remains constant at 16.3%, whereas $Fe_2O_3$ concentrations fluctuate between 4.4 and 5.6%. The contents of CaO and $K_2O$ show variations ranging from 0.6 to 1.1% and 4.7 to 4.8%, respectively. Regarding basalts, notable variations are also observed in the proportions of major elements. The $SiO_2$ content ranges from 40.9% to 41.3%, while $TiO_2$ content is between 4.2% and 4.3%. Similarly, the percentage of $Al_2O_3$

fluctuates between 13.2% and 13.4%. $Fe_2O_3$ concentrations vary slightly, ranging from 15.3% to 15.4%, while CaO content is between 10.1% and 10.4%, and $K_2O$ ranges from 1.4% to 1.7%. These subtle yet significant differences highlight the important geochemical variations between trachyte and basalt samples, underscoring the diversity of their mineralogical composition.



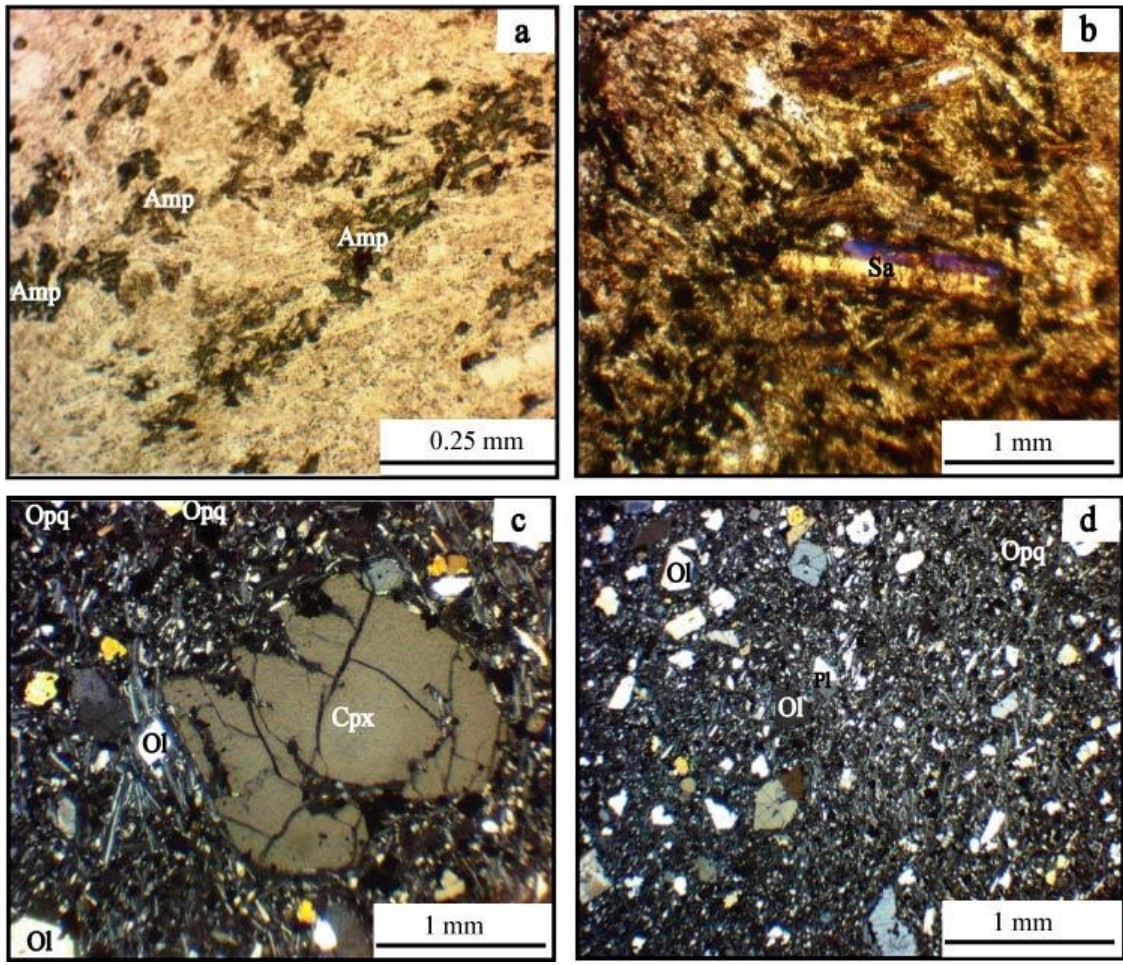

**Fig. 4.** Photomicrographs of thin sections of representative samples from the study area. **(a)** and **(b)** Aphyric texture of Minawao trachyte
showing mineral assemblages. **(c)** and **(d)** Porphyritic texture of Gawar basalt showing mineral assemblages. Ol: olivine. CPx: clinopyroxene.
Pl: plagioclase. Opq: opaque minerals. Sa: sanidine. Amp: amphibole. Mineral abbreviations from Whitney and Evans (2010).

Trace element concentrations in the trachyte show significant fluctuation. The chromium (Cr) content remains relatively
constant at 2 ppm, while other elements such as nickel (Ni), strontium (Sr), rubidium (Rb), yttrium (Y), barium (Ba), and
niobium (Nb) exhibit more pronounced variations, ranging from 7.2 to 9.1 ppm, 7.3 to 20.9 ppm, 136.2 to 148.8 ppm, 60.4 to
97.1 ppm, 16.2 to 48.9 ppm, and 194.2 to 197.3 ppm, respectively. In contrast, basalts show much higher concentrations of Cr
(176.1 to 183 ppm), Ni (269.6 to 277.7 ppm), Sr (1648.5 to 1685.5 ppm), and Ba (561.1 to 572.7 ppm). However, the basalts
have relatively low levels of Rb (18.9 to 23.5 ppm) and Y (22.9 to 23.2 ppm) compared to the trachyte (Table 4). These
variations indicate substantial differences in trace elements between trachytes and basalts, reflecting distinct geochemical
signatures.





***Table 4.*** *Major (wt.%) and trace (mg kg⁻¹) elements data for the Gawar-Zamay lavas.*

| 2022 | Basalte | | | Trachyte | | |
|---|---|---|---|---|---|---|
| | Gaw1 | Gaw2 | Gaw3 | Zam1 | Zam2 | Zam3 |
| Longitude | 13°49'57.4"E | 13°49'55.8"E | 13°49'58.2"E | 13°53'15.9"E | 13°53'17.2"E | 13°53'15,7"E |
| Latitude | 10°30'39.1"N | 10°30'37.3"N | 10°30'38.6"N | 10°36'25.5"N | 10°36'26.8"N | 10°36'26,0"N |
| Altitude (m.a.s.l) | 754 | 751 | 753 | 602 | 604 | 605 |
| (wt%) | | | | | | |
| $SiO_2$ | 41.3 | 41.3 | 40.9 | 66.5 | 66.5 | 65.3 |
| $TiO_2$ | 4.3 | 4.3 | 4.2 | 0.2 | 0.2 | 0.3 |
| $Al2O_3$ | 13.4 | 13.4 | 13.2 | 16.3 | 16.3 | 16.3 |
| $Fe_2O_3$ | 15.3 | 15.3 | 15.4 | 4.4 | 4.4 | 5,6 |
| MnO | 0.2 | 0.2 | 0.2 | 0.1 | 0.1 | 0.2 |
| MgO | 8.5 | 8.5 | 8.7 | 0.1 | 0.0 | 0.0 |
| CaO | 10.4 | 10.5 | 10.1 | 0.7 | 0.7 | 1.1 |
| $Na_2O$ | 3.5 | 3.2 | 3.4 | 7.1 | 7.1 | 6.1 |
| $K_2O$ | 1.4 | 1.7 | 1.7 | 4.7 | 4.8 | 4.8 |
| $P_2O_5$ | 1.1 | 1.1 | 1.1 | 0.0 | 0.0 | 0.0 |
| Total | 99.4 | 99.5 | 98.9 | 99.9 | 99 | 99 |
| Mg | 5.5 | 55.6 | 5,8 | | | |
| (mg kg⁻¹) | | | | | | |
| Sc | 24 | 21.6 | 23.5 | 3 | b.d.l. | 0.9 |
| V | 224.1 | 222.1 | 223 | b.d.l. | b.d.l. | b.d.l |
| Cr | 178 | 176.1 | 183 | 2 | 2 | 2 |
| Co | 60.3 | 59.8 | 62.1 | 7.9 | 7.2 | 9.1 |
| Ni | 271 | 269.6 | 271.7 | 2.4 | 2.6 | 1.4 |
| Ga | 17 | 18.1 | 17.8 | 39.9 | 41.7 | 36.5 |
| Zn | 107.7 | 107.8 | 104.1 | 247.6 | 250.5 | 208.5 |
| Sr | 1668.6 | 1685.5 | 1648.5 | 10.8 | 7.13 | 20.9 |
| Rb | 18.9 | 23.5 | 22.2 | 148.3 | 148.8 | 136.2 |
| Y | 23.2 | 23 | 22.9 | 97.1 | 91.2 | 60.4 |
| Cs | 0.2 | 0.2 | 0.2 | 0.8 | 0.8 | 0.3 |
| Ba | 572.7 | 561.1 | 562.5 | 48.3 | 28.9 | 16.1 |
| La | 49.2 | 48.3 | 48.1 | 168.5 | 160.7 | 184.8 |
| Ce | 106.7 | 104.9 | 104.7 | 420.3 | 413.4 | 477.2 |
| Pr | 13 | 12.9 | 12.8 | 34.7 | 33.4 | 38.1 |
| Nd | 56.9 | 55.9 | 55.8 | 125.4 | 120.8 | 141.4 |
| Sm | 11 | 10.8 | 10.8 | 21.7 | 21.2 | 22.0 |
| Eu | 3.8 | 3.7 | 3.7 | 1.9 | 1.8 | 1.9 |
| Gd | 10.7 | 10.5 | 10.4 | 22.9 | 22.1 | 22.7 |
| Tb | 1.3 | 1.3 | 1.3 | 3 | 2.9 | 2.6 |
| Dy | 6.4 | 6.3 | 6.3 | 16.5 | 15.8 | 12.6 |
| Ho | 1.1 | 1.1 | 1.1 | 3.1 | 3.0 | 2.3 |
| Er | 2.7 | 2.6 | 2.6 | 8.4 | 8.1 | 6.0 |
| Tm | 0.3 | 0.3 | 0.3 | 1.1 | 1.1 | 0.8 |
| Yb | 1.9 | 1.8 | 1.8 | 709 | 6.9 | 5.0 |
| Lu | 0.3 | 0.3 | 0.3 | 1 | 1.0 | 0.7 |
| Hf | 3.3 | 3.6 | 2.9 | 5.4 | 33.5 | 33.4 |
| Ta | 3.5 | 3.5 | 3.4 | 1.9 | 10.8 | 10.6 |
| Pb | 1.8 | 1.7 | 1.7 | 10.5 | 10.4 | 7.8 |



| | | | | | | |
|---|---|---|---|---|---|---|
| Th | 3.8 | 3.7 | 3.7 | 18.6 | 18.3 | 12.1 |
| U | 1.1 | 1.1 | 1.1 | 5.0 | 4.9 | 1.6 |
| Zr | 143.8 | 147.73 | 125.7 | 216 | 1333.5 | 1314.2 |
| Nb | 67.1 | 67.67 | 67.2 | 33.6 | 197.3 | 194.2 |

b.d.l: below detection limit

## 3.2 Morphological and mineralogical characteristics of the soil

The studied soil profile consists of six horizons (Fig. 5.) divided into four distinct textural groups: sandy loam, loamy sand, loamy clayey sand, clayey sand and sandy. The colours range from light yellow-brown (10YR 5/4) to dark brown (7.5YR 3/2), passing through light yellow (10YR 5/4) and up to pale yellow (2.5Y 7/4). Each horizon has a fine, granular structure with significant biological and matrix porosity, promoting good permeability and optimal friability. The transitions between horizons are gradual and regular (Table 5).

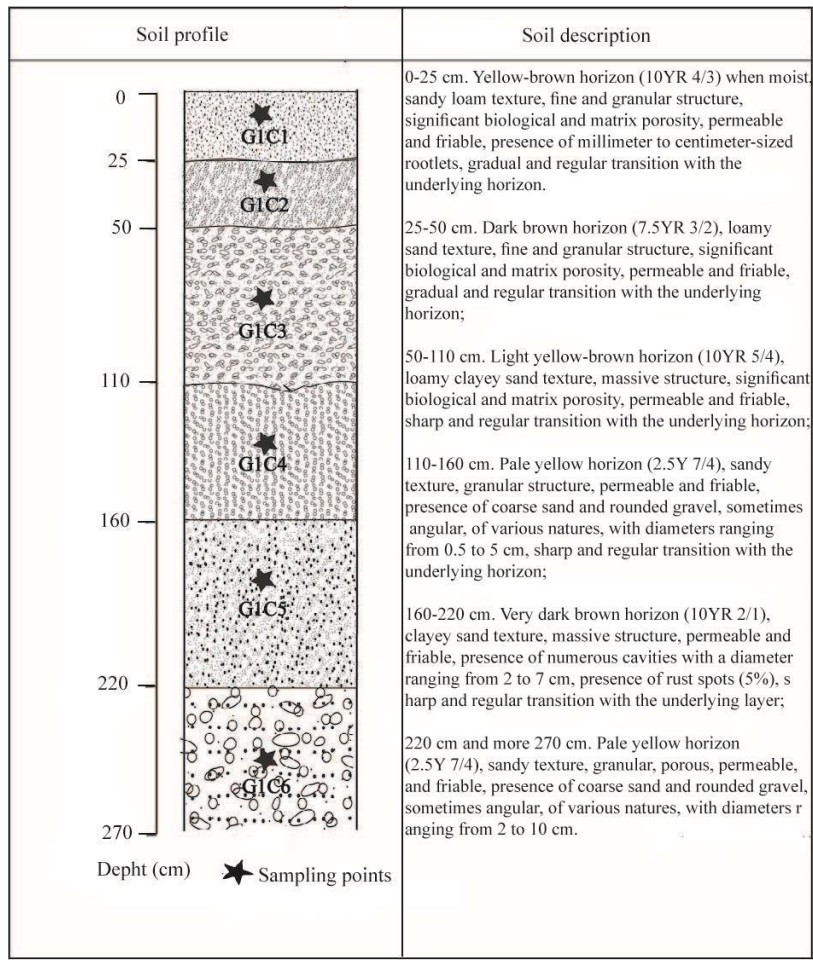

**Fig. 5.** Macroscopic organization of the soil in the study area.




*Table 5. Morphological and physical characteristics of the soil profile.*

| Site | Depth(cm) | Colour (Moist) | Structure | Consistency Dry | Wet | Rock fragments | Boundary | Textural class |
|------|-----------|----------------|-----------|-----|-----|----------------|----------|----------------|
| Guiring | 0-25 | 10YR 4/3 | f&wg | s | s & p | n | g | sandy loam |
| | 25-50 | 7.5YR 3/2 | f&wm | s | s & p | c | g | loamy sand |
| | 50-110 | 10YR 5/4 | m&mma | h | s & p | m | g | loamy clayey sand |
| | 110-160 | 2.5Y 7/4 | c&sabk | l | s & p | d | c | sandy |
| | 160-220 | 10YR 2/1 | m&sma | h | s & p | m | c | loamy clayey sand |
| | 220-270 | 2.5Y 7/4 | c&sabk | l | s & p | d | g | sandy |

Soil characteristics

| Structure | | | Consistency | | Rock fragments | Boundary |
|-----------|---|---|-------------|---|----------------|----------|
| Size | Type | Grade | Dry | Wet | | |
| vf = very fine (G5 mm) | g = granular | w = weak (peds barely observable) | l = loose | s = sticky | n = none (0%) | g = gradual |
| f = fine (5–10 mm) | abk = angular blocky | | | p = plastic | m = many (15–40%) | c = clear |
| m = medium (10–20 mm) | sbk = subangular blocky | m = moderate (peds observable) | s = soft | | v = very few (0–2%) | |
| c = coarse (20–50 mm) | | | h = hard | | a = abundant (40–80%) | |
| vc = very coarse (>50 mm) | l=lumpy | s = strong (peds clearly observable) | | | c = common (5–15%) | |
| | ma=massive | | | | d = dominant (>80%) | |

Mineralogically, the infrared spectra of the soil, illustrated in Fig. 6, reveal the presence of kaolinite, smectites, sepiolite, and quartz. In the high-frequency region, the hydroxyl groups display two bands at 3774 – 3622 cm⁻¹, corresponding to the stretching vibrations of the OH⁻ groups characteristic of kaolinite according to Famer (1964) and Frost et al. (1998). The low intensity absorption band located at 1621 cm⁻¹ is attributed to water molecules, possibly associated with smectites. Sepiolite is primarily distinguished by the band at 531 cm⁻¹. The presence of quartz is manifested by the peaks at 1003, 774, 693, 531, and 459 cm⁻¹, which correspond to Si-O vibrations.

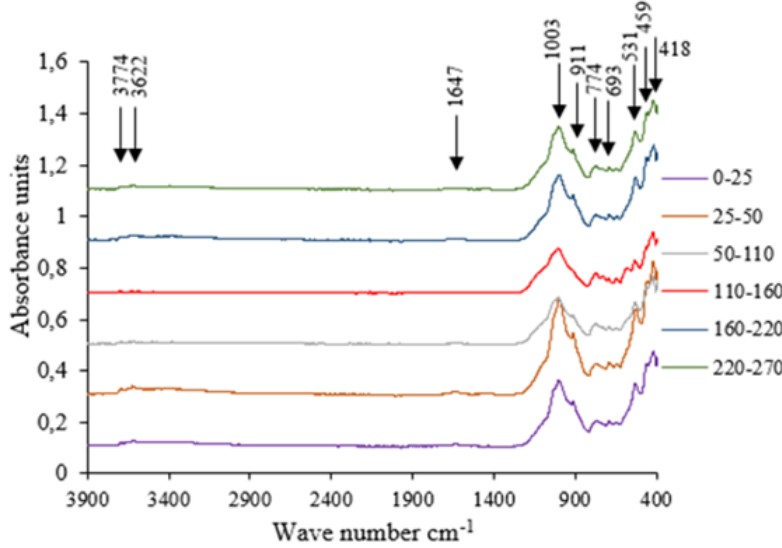

**Fig. 6.** Infrared spectra of the studied soil.



### 3.3 Physicochemical characteristics of the soils

The particle size distribution is dominated by sand fraction with proportions varying from 62% to 82%, corresponding to a mean value of $76.2 \pm 9.2\%$ (Table 6). Clay constitutes the second important particle size fraction, with proportions ranging

from 13% to 23%, with a mean value of $16.3 \pm 5.2$ %. Silt fraction appears to be the least abundant, with proportions varying from 5% to 15%, with an average of $9.2 \pm 4.9\%$.

The analysed soils exhibit a pH ranging from acidic to slightly basic (6.8 to 7.2), with an average of $7.0 \pm 0.2$. No correlation was observed between pH and other elements, except for clay ($r = 0.9$, $p < 0.05$). The organic matter (OM) and total nitrogen (N) contents are relatively low, with values ranging from 1.3% to 3.2%, and 0.1%, with averages of $2.6 \pm 0.7\%$, and $1.1 \pm$

0.0% (Table 6). The C/N ratio, relatively low, varies from 6 to 16, with an average of $12.3 \pm 3.6$. Base saturation (BS) varies between 23.6% and 42.4%, with an average value of $31.1 \pm 7.4\%$ (Table 6).

The total exchangeable bases ($Ca^{2+}$, $Mg^{2+}$, $K^+$, and $Na^+$) exhibit notable variation across the soil profile. Calcium ($Ca^{2+}$) and magnesium ($Mg^{2+}$) concentrations are low to moderate, with values ranging from 2.6 to 6.0 $cmol_c\ kg^{-1}$ in the case of $Ca^{2+}$ and 0.7 to 5.4 $cmol_c\ kg^{-1}$ for $Mg^{2+}$, with respective averages of $3.9 \pm 1.5$ and $2.5 \pm 1.6$ $cmol_c\ kg^{-1}$ (Table 6). Regarding potassium

($K^+$) and sodium ($Na^+$), their contents are very low to low, oscillating between 0.1 and 0.4 $cmol_c\ kg^{-1}$ for $K^+$ and between 0.1 and 0.6 $cmol_c\ kg^{-1}$ for $Na^+$, with respective averages of $0.2 \pm 0.1$ and $0.3 \pm 0.2$ $cmol_c\ kg^{-1}$ (Table 6). $Mg^{2+}$ shows a positive correlation with clay ($r = 0.6$, $p < 0.05$) and pH $H_2O$ ($r = 0.7$, $p < 0.05$). $K^+$ is strongly correlated with clay ($r = 0.8$, $p < 0.05$) and moderately with pH $H_2O$ ($r = 0.7$, $p < 0.05$). The total of exchangeable bases (TEB) ranges from 5.1 to 8.9 $cmol\cdot kg^{-1}$, with an average value of $6.9 \pm 1.7$ $cmol_c\ kg^{-1}$ (Table 6). This TEB shows a moderately positive correlation with clay content ($r =$

$0.8$, $p < 0.05$). The cation exchange capacity (CEC) displays relatively high values, ranging from 18.7 to 25.0 $cmol_c\ kg^{-1}$, with an average of $22.1 \pm 2.5$ $cmol_c\ kg^{-1}$ (Table 6). It shows a strong correlation with the clay fraction ($r = 0.8$, $p < 0.05$), as well as with pH $H_2O$ ($r = 0.8$, $p < 0.05$). It is also moderately correlated with potassium ($K^+$) ($r = 0.8$, $p < 0.05$) (Table 7). Phosphorus (P) shows concentrations ranging from low to medium, with values between 12.6 and 30.3 $mg\ kg^{-1}$, and an average of $19.0 \pm$ 7.0 $mg\ kg^{-1}$ (Table 6). Phosphorus shows a strong correlation with clay content ($r = 0.9$, $p < 0.05$) and with CEC ($r = 0.8$, $p <$

$0.05$) (Table 7). Base saturation (BS) presents values ranging from 23.6% to 42.4%, with an average of $31.1 \pm 7.4\%$. Electrical conductivity varies from 0.0 to 1.2 dS/m, with an average of $0.3 \pm 0.5$ dS/m (Table 6). The latter is strongly correlated with Na, with a coefficient of $r = 0.9$ ($p < 0.05$).

Morphologically, the study soil profile is characterized by an alternation of different soil texture which are sandy loam, loamy sand, loamy clayey sand, clayey sand and sandy, in line with the alluvial nature of the parent rock. This alternation of soil

texture is confirmed by the zigzag evolution of the percentage of different particle size distribution fraction noted in table 5. These characteristics correspond to those of Fluvisols. Base saturation ratio in all soil horizons is below 50%, leading to the choice of Dystric as principal qualifier. The light colour of soil and the organic matter content (<3%) lead to classify the studied soils as Ochric Dystric Fluvisols according to the IUSS Working Group WRB (2022).



*Table 6. Physicochemical characteristics and summary statistics of the soil profile.*

| Depth cm / statistics | 0-25 | 25-50 | 50-110 | 110-160 | 160-220 | 220-270 | Min | Max | Mean | SD | CV |
|---|---|---|---|---|---|---|---|---|---|---|---|
| Sand (%) | 82.0 | 82.0 | 67.0 | 88.0 | 62.0 | 88.0 | 62.0 | 82.0 | 76.2 | 9.2 | 0.1 |
| Silt (%) | 15.0 | 5.0 | 10.0 | 8.0 | 15.0 | 8.0 | 5.0 | 15.0 | 9.2 | 4.9 | 0.5 |
| Clay (%) | 13.0 | 13.0 | 23.0 | 5.0 | 23.0 | 5.0 | 5.0 | 23.0 | 16.3 | 5.2 | 0.3 |
| $pH_{H2O}$ | 6.8 | 7.0 | 7.2 | 6.8 | 7.2 | 6.9 | 6.8 | 7.2 | 7.0 | 0.2 | 0.0 |
| $pH_{KCl}$ | 4.4 | 4.2 | 6.3 | 6.6 | 4.6 | 5.5 | 4.2 | 6.6 | 5.3 | 1.0 | 0.2 |
| OC (%) | 0.8 | 1.8 | 1.7 | 1.7 | 1.4 | 1.7 | 0.8 | 1.8 | 1.5 | 0.4 | 0.3 |
| OM (%) | 1.3 | 3.2 | 2.9 | 2.9 | 2.3 | 2.9 | 1.3 | 3.2 | 2.6 | 0.7 | 0.3 |
| N (%) | 0.1 | 0.1 | 0.1 | 0.1 | 0.1 | 0.1 | 0.1 | 0.1 | 0.1 | 0.0 | 0.1 |
| C/N | 6.0 | 14.0 | 16.0 | 12.0 | 11.0 | 15.0 | 6.0 | 16.0 | 12.3 | 3.6 | 0.3 |
| Ca ($cmol_c\ kg^{-1}$) | 3.0 | 2.6 | 2.7 | 5.5 | 6.0 | 3.4 | 2.6 | 6.0 | 3.9 | 1.5 | 0.4 |
| Mg ($cmol_c\ kg^{-1}$) | 2.2 | 2.5 | 5.4 | 0.7 | 2.0 | 1.8 | 0.7 | 5.4 | 2.5 | 1.6 | 0.7 |
| K ($cmol_c\ kg^{-1}$) | 0.2 | 0.1 | 0.4 | 0.1 | 0.2 | 0.1 | 0.1 | 0.4 | 0.2 | 0.1 | 0.6 |
| Na ($cmol_c\ kg^{-1}$) | 0.2 | 0.2 | 0.3 | 0.6 | 0.1 | 0.3 | 0.1 | 0.6 | 0.3 | 0.2 | 0.6 |
| TEB ($cmol_c\ kg^{-1}$) | 5.6 | 5.5 | 8.9 | 7.9 | 8.3 | 5.1 | 5.1 | 8.9 | 6.9 | 1.7 | 0.2 |
| CEC ($cmol_c\ kg^{-1}$) | 20.8 | 23.2 | 25.0 | 18.7 | 24.4 | 20.2 | 18.7 | 25.0 | 22.1 | 2.5 | 0.1 |
| P ($mg\ kg^{-1}$) | 13.1 | 12.6 | 24.3 | 17.5 | 30.3 | 16.2 | 12.6 | 30.3 | 19.0 | 7.0 | 0.4 |
| BS (%) | 26.8 | 23.6 | 34.5 | 42.4 | 34.1 | 25.3 | 23.6 | 42.4 | 31.1 | 7.4 | 23.5 |
| CE ($mS\ cm^{-1}$) | 0.1 | 0.0 | 0.2 | 1.2 | 0.0 | 0.1 | 0.0 | 1.2 | 0.3 | 0.5 | 1.7 |


*Table 7. Spearman correlation matrix for linear relationships between soil profile properties.*

| | Sand | Silt | Clay | $pH_{H2O}$ | $pH_{KCl}$ | OC | OM | N | C/N | Ca | Mg) | K | Na | TEB | CEC | P | BS | CE |
|---|---|---|---|---|---|---|---|---|---|---|---|---|---|---|---|---|---|---|
| Sand | 1 | | | | | | | | | | | | | | | | | |
| Silt | -0.6 | 1 | | | | | | | | | | | | | | | | |
| Clay | -0.9* | 0.5 | 1 | | | | | | | | | | | | | | | |
| $pH_{H2O}$ | -0.9* | 0.3 | 0.9 | 1 | | | | | | | | | | | | | | |
| $pH_{KCl}$ | -0.1 | -0.4 | 0.1 | -0.1 | 1 | | | | | | | | | | | | | |
| OC | 0.0 | -0.8* | 0.0 | 0.3 | 0.4 | 1 | | | | | | | | | | | | |
| OM | 0.0 | -0.8* | 0.0 | 0.3 | 0.4 | 0.9* | 1 | | | | | | | | | | | |
| N | 0.6 | -0.2 | -0.6 | -0.6 | -0.3 | -0.1 | -0.1 | 1 | | | | | | | | | | |
| C/N | -0.2 | -0.7 | 0.3 | 0.5 | 0.5 | 0.9* | 0.9* | -0.5 | 1 | | | | | | | | | |
| Ca | -0.4 | 0.2 | 0.3 | 0.1 | 0.2 | -0.0 | -0.0 | 0.2 | -0.2 | 1 | | | | | | | | |
| Mg | -0.5 | 0.2 | 0.6 | 0.7 | 0.1 | 0.1 | 0.1 | -0.7 | 0.4 | -0.6 | 1 | | | | | | | |
| K | -0.6 | 0.3 | 0.8 | 0.7 | 0.4 | 0.1 | 0.1 | -0.7 | 0.4 | -0.2 | 0.9* | 1 | | | | | | |
| Na | 0.4 | -0.6 | -0.4 | -0.5 | 0.8 | 0.4 | 0.4 | 0.2 | 0.2 | 0.2 | -0.3 | -0.0 | 1 | | | | | |
| TEB | -0.8 | 0.3 | 0.8 | 0.6 | 0.6 | 0.2 | 0.2 | -0.3 | 0.2 | 0.5 | 0.4 | 0.7 | 0.2 | 1 | | | | |
| CEC | -0.8 | 0.4 | 0.8* | 0.9* | -0.3 | 0.1 | 0.1 | -0.5 | 0.3 | -0.2 | 0.8 | 0.7 | -0.7 | 0.4 | 1 | | | |
| P | -0.9* | 0.5 | 0.9* | 0.8* | 0.2 | 0.1 | 0.1 | -0.5 | 0.2 | 0.6 | 0.3 | 0.5 | -0.3 | 0.8* | 0.6 | 1 | | |
| BS | -0.3 | 0.0 | 0.4 | 0.1 | 0.7 | 0.1 | 0.1 | 0.0 | 0.0 | 0.7 | -0.1 | 0.3 | 0.6 | 0.8* | -0.1 | 0.5 | 1 | |
| CE | 0.3 | -0.4 | -0.3 | -0.5 | 0.7 | 0.2 | 0.2 | 0.4 | 0.0 | 0.5 | -0.4 | -0.1 | 0.9 | 0.4 | -0.6 | -0.1 | 0.8* | 1 |

**\*Significant at p < 0.05.



### 3.4 Suitability of the studied soil for maize cultivation

The study area presents very favourable climatic conditions for the growth and yield of maize. The average temperature of 26.6 °C during the growing cycle presents a slight limitation for the production of maize (S1-1) (Table 8). The precipitation during the growing cycle, approximately 649 mm, has no limitation for the cultivation of maize. The climatic index of 91.4 (S1-0) indicates that there are no major limitations for maize cultivation in this area. This means that climatic conditions are favourable, suggesting the possibility of achieving optimal yields.

**Table 8.** *Climatic suitability evaluation for the production of Maize using simple limitation and parametric methods.*

| Characteristics | Values | Classes | Limitations | Parametric values |
|---|---|---|---|---|
| **Precipitation** | | | | |
| Precipitation during crop cycle (mm) | 649.2 | S1-0 | 0 | 100 |
| Precipitation during 3rd cycle (mm) | 232.4 | S1-0 | 0 | 100 |
| **Temperature** | | | | |
| Mean temperature during crop cycle (°C) | 26.6 | S1-1 | 1 | 95 |
| Mean maximum temperature during crop cycle (°C) | 31.8 | S1-1 | 1 | 86.9 |
| Mean minimum temperature during crop cycle (°C) | 23 | S1-1 | 1 | 93 |
| Relative humidity during crop cycle (%) | 73.3 | S1-0 | 0 | 100 |
| n/N during crop cycle | 62.6 | S1-1 | 0 | 100 |
| Calculated climatic index (CR) | | S1-1 | | 91.4 |

The low slope (1%), presents a slight limitation (S1-1). The risk of flooding is non-existent (FO), and drainage is of good quality, indicating that there are no constraints related to soil moisture (S1-0) (Table 9). The soil texture, of the sandy-loamy type (SL), classified as S2, has proven to be a moderate limitation with a parametric value of 67.4. The soil depth, greater than 100 cm, is very suitable (S1-0), which is optimal for root development. Presence of coarse fragments displays a slight limitation (S1.1), with a parametric value of 96.3 for maize production (Table 9). The apparent cation exchange capacity (CEC) of the soil, classified as S1-1, indicates a slight limitation, with a parametric value of 90. Base saturation and organic carbon content, both classified as S2, reveal moderate limitations, with respective parametric values of 75.2 and 72.8. The soil pH (S1-1) and the electrical conductivity is suitable (S1-0) for the production of maize although there is a slight limitation related to pH (Table 9). The overall land suitability index is 62.4, which classifies it as S2sf, indicating moderate suitability for cultivation, primarily due to constraints related to the soil's physical properties and fertility.




*Table 9. Land suitability evaluation for the maize production using simple limitation and parametric methods.*

| Characteristics | Values | Classes | Limitations | Parametric values |
|---|---|---|---|---|
| **Topography (t)** | | | | |
| Slope (%) | 1 | S1-1 | 1 | 93.2 |
| Wetness (w) | | | | |
| Flooding | FO | S1-0 | 0 | 100 |
| Drainage | Good | S1-0 | 0 | 100 |
| **Physical soil characteristics (s)** | | | | |
| Texture/structure | SL | S2 | 2 | 67.4 |
| Soil depth (cm) | > 100 | S1-0 | 0 | 100 |
| Coarse fragments (%) | | S1-1 | 1 | 96.3 |
| **Soil fertility characteristics (f)** | | | | |
| Apparent CEC ($cmol_c$ $kg^{-1}$ clay) | 22.1 | S1-1 | 1 | 90 |
| Base saturation (%) | 31.3 | S2 | 2 | 75.2 |
| Organic carbon (%) | 0.8 | S2 | 2 | 72.8 |
| pH-$H_2O$ | 7 | S1-1 | 1 | 94 |
| CE | 0.3 | S1-0 | 0 | 100 |
| Suitability and calculated land index (Is) | | S2sf | | 62.4 |

## 3.5 Evaluation of the effect of trachyte and basalt powder on the growth and yield of maize

### 3.5.1 Effect of trachyte and basalt powder on maize growth parameters

The maize growth parameters include the germination rate, plant height, number of leaves, stem diameter, as well as ear length
and ear diameter.

Germination was observed in all plots starting from the 5th day after sowing. After 10 days, the recorded germination rate
reached 92.2%, indicating good uniformity and a of plant growth across all experimental plots.

The plant height ranges from 126.5±14.5 cm for the control plot without treatment ($T_0$) to 239.2±19.7 cm for treatment with
NPK fertilizer + urea ($T_5$), with the latter showing the tallest plants, while $T_0$ shows the shortest height. Treatments with
trachyte powder ($T_1$: 152.1±10.4 cm), trachyte powder + urea ($T_2$: 171.5±3.0 cm), basalt powder ($T_3$: 159.1±6.4 cm), and basalt
powder + urea ($T_4$: 177.6±10.4 cm) show significant differences among them, and all are higher than $T_0$. However, treatment
$T_5$, which is a reference fertilization practice in the region (239.2±19.7 cm) is significantly ($p < 0.05$) taller than all other
treatments. A progressive increase in plant height is observed with inputs of trachyte and basalt powders (Table 10).

The evolution of the number of leaves based on treatments shows that treatment $T_5$, with an average of 22 leaves per plant
(22±0.2), yields the best results, showing a significant difference ($p < 0.05$) compared to the other treatments. In contrast,
treatment $T_0$, with 18 leaves per plant (18±0.4), recorded the lowest number of leaves. Treatments $T_1$ and $T_3$ produced similar





results, with 19 leaves per plant, slightly lower than that of $T_2$ and $T_4$, with an average of 20 leaves per plant each. There is a significant difference ($p < 0.05$) between T3 and T4, and between T0 and T5 and the other treatments.

The stem diameter shows a significant ($p < 0.05$) increase depending on the treatments (Table 9). $T_0$ recorded the smallest stem
diameter ($1.4\pm0.2$ cm), serving as the reference without any treatment influence. $T_1$ and $T_2$ had stem diameters of $2.3\pm0.1$ cm and $2.7\pm0.1$ cm, respectively, both larger than $T_0$. Similarly, $T_3$ recorded a stem diameter of $2.6\pm0.2$ cm, and $T_4$ reached $2.7\pm0.1$ cm, both showing wider diameters than $T_0$. $T_5$ exhibited the largest diameter at $3.0\pm0.1$ cm. The plots treated with trachyte and basalt powder clearly stand out from the control plots ($T_0$).

The evolution of ear length shows a significant ($p < 0.05$) increase as the treatments are applied (Table 10). Treatment $T_0$, with
an ear length of $8.4\pm0.4$ cm, presents the shortest ears and stands out significantly ($p < 0.05$) from all other treatments. $T_1$ ($12.2\pm0.4$ cm) and $T_2$ ($13.1\pm0.1$ cm) display noticeably longer ears than those of $T_0$. For treatments $T_3$ and $T_4$, the ear length increases to $12.8\pm0.1$ cm and $14.1\pm0.7$ cm, respectively. Under the influence of treatment $T_5$ known as reference fertilization practice in the region, the length reaches its maximum at $15.1\pm0.2$ cm, a value significantly ($p < 0.05$) higher than the others (Table 10).

The ear diameter shows a statistically significant ($p < 0.05$) increase with the application of the different treatments (Table 10). Treatment $T_0$, with an average diameter of $3.0\pm0.2$ cm, produces the smallest ears. $T_1$ ($3.6\pm0.1$ cm) and $T_2$ ($3.9\pm0.3$ cm) show a noticeable increase compared to $T_0$. Treatment $T_3$ ($4.1\pm0.0$) and $T_4$, with a diameter of $4.2\pm0.1$ cm, also show significant ($p < 0.05$) increases, surpassing $T_0$. Under the effect of treatment $T_5$, the diameter reaches its maximum at $5.0\pm0.2$ cm, a value significantly ($p < 0.05$) higher than those observed for the other treatments.

**3.5.2 Effect of trachyte and basalt powder on maize yield parameters**

Several parameters help assess the overall productivity of maize, including the weight of the ears, the weight of 100 maize grains, and the total yield in kilograms per hectare (kg ha$^{-1}$).

The weight of the ears increases significantly ($p < 0.05$) according to the treatments, indicating a positive response over time, resulting from the dissolution of the rock powder (Table 10). Treatment $T_0$, with a weight of $29.8\pm1.5$ g, is significantly
($p < 0.05$) lower than all the others. Under treatment $T_1$, the weight doubles, reaching $77.6\pm4.6$ g, and $T_2$ ($93.5\pm4.6$ g) also shows a notable improvement compared to $T_0$. Treatment $T_3$ ($81.7\pm3.4$ g) and $T_4$ ($97.5\pm2.4$ g), are statistically significant ($p < 0.05$) compared to $T_0$. Treatment $T_5$, with a weight of $128.3\pm2.4$ g, shows a substantial and significant ($p < 0.05$) increase compared to all other treatments.

The variation in the weight of 100 grains shows that treatment $T_0$, with a weight of $18.8\pm1.5$ g, is significantly ($p < 0.05$) lower
than all the other treatments. Treatments $T_1$ ($23.7\pm1.4$ g) and $T_2$ ($27.9\pm1.7$ g) exhibit notable increases compared to $T_0$, Treatments $T_3$ ($26.1\pm1.2$ g) and $T_4$ ($28.7\pm1.6$) also show significantly ($p < 0.05$) higher values than $T_0$. Treatment $T_5$, with an average weight of $32.1\pm2.5$ g for 100 grains, significantly ($p < 0.05$) outperforms all other treatments (Table 10).





The yield in kg ha$^{-1}$ based on the treatments is presented in Table 10. Treatment T$_0$, with a yield of 645.8±27.0 kg ha$^{-1}$, shows the lowest level among all treatments. In contrast, treatments T$_1$ (2362.9±27.0 kg ha$^{-1}$) and T$_2$ (2763.9±27.0 kg ha$^{-1}$) recorded

significantly ($p < 0.05$) higher yields, demonstrating notable improvement compared to T$_0$ due to the application of trachyte powder. Similarly, treatment T$_3$ (2558.6±27.0 kg ha$^{-1}$) and treatment T$_4$ (2931.2±27.0 kg ha$^{-1}$) reveal significant ($p < 0.05$) increases, highlighting the positive impact of basalt powder on maize yield. Treatment T$_5$, with a maximum yield of 3164.5±27.0 kg ha$^{-1}$, statistically outperforms all other treatments. Globally, it is noted that application of basalt powder has more positive effect on maize yields than trachyte powder while adding urea to these treatments increase the maize yield.

***Table 10.*** *Statistical analysis of the effect of basalt and trachyte powder on the growth and yield of Maize parameters (Mean ± SD\*).*

| Variable | T$_0$ | T$_1$ | T$_2$ | T$_3$ | T$_4$ | T$_5$ |
|---|---|---|---|---|---|---|
| **Plant height (cm)** | 126.5±14.5a | 152.1±10.4b | 171.5±3.0c | 159.1±6.4b | 177.6±10.4d | 239.2±19.7e |
| **Number of leaves** | 18±0.4a | 19±0.1b | 20±0.1b | 19±0.3c | 20±0.3b | 22±0.2d |
| **Stem diameter (cm)** | 1.4±0.2a | 2.34±0.1b | 2.7±0.1c | 2.6±0.2c | 2.7±0.7d | 3.5±0.1e |
| **Ear length (cm)** | 8.4±0.4a | 12.2±0.4b | 13.1±0.1c | 12.8±0.1cd | 14.1±0.1ab | 15.1±0.2b |
| **Ear diameter (cm)** | 3.0±0.2a | 3.6±0.1b | 3.9±0.3c | 4.1±0.0c | 4.2±0.1d | 5.0±0.2e |
| **Ear weight (g)** | 29.8±1.5a | 77.6±4.6b | 93.4±4.6c | 81.7±3.4d | 97.2±2.4e | 128.3±2.4f |
| **Weight of 100 grains (g)** | 18.8±1.5a | 23.7±1.4b | 27.9±1.7c | 26.1±1.2cd | 28.7±1.6cd | 32.1±2.5d |
| **Yield (kg ha$^{-1}$)** | 645.8±27.0a | 2362.9±27.0b | 2763.9±27.0c | 2558.6±27.0d | 2931.2±27.0e | 3164.5±27.0f |

T$_0$: Control plot without treatment; T$_1$: Treatment with trachyte powder (1.96 kg of trachyte powder for 9.6m$^2$); T$_2$: Treatment with trachyte powder (2.0 kg of trachyte powder for 9.6m$^2$) + urea (100g for 9.6m$^2$) at 06 weeks; T$_3$: Treatment with basalt powder (2.0 kg of basalt powder for 9.6m$^2$); T$_4$: Treatment with basalt powder (2.0 kg of trachyte powder for 9.6m$^2$) + urea (100g for 9.6 m$^2$) at 06 weeks. T$_5$: Treatment with NPK fertilizer (200g for 9.6 m$^2$) + urea (100g for 9.6 m$^2$) at 06 weeks; * Standard deviation; Mean value followed by different lower-case

letters within the same line have significant differences ($p < 0.05$)

## 4 Discussion

### 4.1 Physical and mineralogical characteristics of soils

Particle size analyses revealed that the upper part of the soil profile primarily consists of a sandy-loam horizon, characterized

by a high sand content ranging from 67 to 82%. The underlying horizons exhibit a succession of sandy and sandy-clay textures, with similar sand proportions. These observations are consistent with those observed in Fluvisols from various arid regions around the world (Romanens et al., 2019). The predominance of sand fractions can negatively affect water and nutrient retention capacities while directly impacting plant root development (Basga et al., 2018; Tsozué et al., 2020a). The limited presence of silt particles may help mitigate these effects by enhancing infiltration and slightly increasing the soil's water and



nutrient retention capacity (Lu et al., 2020; Tsozué et al., 2021). The clay fraction, ranging from 13% to 18%, plays a significant role in the soil's structural stability and fertility, and can enhance both nutrient availability and water retention in the soil (Kome et al., 2019; Gautam et al., 2022).

From a mineralogical perspective, the studied soils are predominantly composed of quartz, along with clay minerals such as kaolinite, smectites, and sepiolite. Kaolinite, an indicator of advanced weathering processes in tropical environments, results

from climatic conditions that promote the leaching of basic cations and the neoformation of secondary minerals (Tsozué et al., 2020b; Lyu and Lu, 2024). In contrast, the presence of smectite suggests moderate pedoclimatic conditions with variable humidity, which enhance cation exchange capacity and water retention. However, due to its swelling and shrinking behaviour, smectite could cause drainage problems and surface water stagnation, limiting agricultural use for certain sensitive crops (Hopmans et al., 2021). Sepiolite, on the other hand, is characteristic of semi-arid environments with low leaching rates

(Bannari et al., 2021). According to Easwaran et al. (2024), this mineral is notable for its remarkable ability to retain water and nutrients, making it particularly beneficial for soils in arid areas or regions prone to drought.

## 4.2 Physicochemical characteristics of the studied soils

From a physicochemical perspective, the pH reveals low acidity, typical of soils under a Sudano-Sahelian climate, consistent with the chemical conditions observed in this region (Tsozué et al., 2020a; Tamto Mamdem et al., 2024). The slight

acidification, particularly in the surface layer, can be attributed to the leaching of bases and the use of synthetic fertilizers and pesticides (Pahalvi et al., 2021; Sarkar et al., 2024). The strong correlation between pH and cation exchange capacity highlights the importance of regulating pH to optimize CEC and, consequently, improve the availability of essential nutrients for crops (Tsozué et al., 2021; Poggi et al. 2023; Santos et al., 2023; Thiaw et al., 2024). Organic matter plays a key role in soil fertility; its low proportion could be explained by several factors, including overexploitation of agricultural lands and intensive grazing

after harvest, the semi-arid climate characteristic of the study area and the sandy texture of the soils (Abdelhak, 2022; Plaza et al., 2018; De Alencar et al., 2023). Improved management practices, such as adding organic matter, crop rotation, or using cover crops to improve soil fertility and overall health, could be corrective measures (Arif et al., 2021; Choudhury et al., 2024; Yang et al., 2024). The very low C/N ratio indicates rapid mineralization, likely related to high temperatures and the sandy texture of the soils (Chen et al., 2024). The low content of available phosphorus in the soil, although essential for crop growth

and yield, could be attributed to its fixation on the clay-humic complex, a phenomenon strongly influenced by the clay content, pH, and calcium levels (Joshi et al., 2021; Lotse Tedontsah et al., 2022). The relatively high CEC is a crucial indicator of soil fertility, illustrating its ability to retain and exchange nutrients, essential for plant growth. According to Obi et al. (2020), this suggests that the soil can sustainably provide nutrients. High concentrations of $Ca^{2+}$ and $Mg^{2+}$, especially at depth, indicate a good reserve of nutrients for plants (Hechmi et al., 2023). In contrast, the relatively low $K^+$ content could limit plant growth,

as potassium is important for photosynthesis and water regulation (Johnson et al., 2022). The moderate $Na^+$ concentration would be beneficial in preventing salinity issues (Hussain et al., 2021). These exchangeable bases play a vital role in plant





health and improving agricultural yields (Verma et al., 2024). Electrical conductivity is a key indicator of soil salinity. Since the measured values are low, this would not favour optimal absorption of water and nutrients by plant roots (Golia, 2023).

**4.3 Kinetics of nutrient element release from trachyte and basalt powders**

The release of nutrients from trachyte and basalt powders is a key topic in pedology and geochemistry, particularly in biogeochemical weathering (Podia et al., 2023). These processes enhance the bioavailability of essential plant nutrients (Medeiros et al., 2023). Mineral hydration from water input causes swelling and disintegration, releasing cations like K, Ca, and Mg (Ramos et al., 2022; Wakeel, 2013). The release of nutrients from trachyte and basalt powders is governed by interdependent physical, chemical, and biological processes (Swoboda et al., 2022). Basalt, rich in plagioclase feldspars,

pyroxenes, and olivine, provides Ca, Mg, Fe, and trace elements. Its high reactivity stems from mafic minerals, which are more easily altered (Korndörfer et al., 2002; Ghasera and Rashid, 2024). In contrast, trachyte, mainly composed of alkali feldspars, is a major K source but has lower reactivity due to its felsic nature (Meena and Biswas, 2014; Mbissik et al., 2021). Particle size significantly influences mineral dissolution and nutrient release (Duarte et al., 2013; Burbano et al., 2022; Rodrigues et al., 2024b). Meena and Biswas (2014) demonstrated that rock phosphate and mica release cations more rapidly when finely

ground and combined with compost. Acidic pH favours silicate dissolution and K, Mg, and Fe release, while reducing conditions increase Fe and Mn mobility (Wakeel, 2013; Vondráčková et al., 2017). Organic matter and microbial activity play key roles in mineral weathering (Rout et al., 2024). *Bacillus mucilaginosus* produces organic acids that enhance nutrient solubilization (Yang et al., 2016). Root exudates and mycorrhizal associations further improve K and Mg bioavailability, facilitating plant uptake (Ma and Yamaji, 2006; Menna and Biswas, 2014). This microbial activity, coupled with mycorrhizal

interactions, enhances mineral breakdown and nutrient uptake (Menna and Biswas, 2014). These symbioses boost root access to nutrients and activate biological processes that promote silicate dissolution (Yadegari et al., 2024). These combined mechanisms make trachyte and basalt powders particularly effective in nutrient-deficient soils, facilitating the release of K, Ca, and Mg. High humidity promotes mineral alteration and microbial activity, while excessive rainfall can cause nutrient leaching (Basak and Biswas, 2009; Lima et al., 2010; Duarte et al., 2013).

**4.4 Suitability of the soils from Guiring to maize production**

     The climate of the study area offers favourable conditions for maize growth and yield, as evidenced by annual rainfall and the climatic index. Conversely, some edaphic parameters pose moderate limitations, including the sandy-loam texture and low base saturation, which affect water retention and nutrient availability (Verdoodt et al., 2003; Tsozué et al., 2015; Issiné et al., 2022; Scanlan et al., 2022). To address these limitations, adapted management practices are essential. Adding organic matter,

such as compost or crop residues, and using cover crops like legumes can improve soil structure and water retention capacity (Obi et al., 2020). Additionally, the application of lime or dolomite could increase calcium and magnesium saturation, while potassium fertilizers can correct low potassium availability (Hechmi et al., 2023). Improving soil texture is also a critical



strategy. Adding clay or biochar, identified by Hussain et al. (2021) as an effective solution, enhances water retention and cation exchange capacity, both of which are fundamental properties for the fertility of sandy soils. The use of organic fertilizers is similarly effective in overcoming fertility limitations while promoting sustainable soil management (Van Leeuwen et al., 2015; Das et al., 2024; Robinson, 2024). Trachyte and basalt powders, used as natural amendments, represent a promising alternative for improving soil fertility, especially in semi-arid regions where soils are often degraded by nutrient depletion and poor structure. These powders gradually release essential mineral elements such as calcium, magnesium, and potassium, thereby increasing exchangeable bases and cation saturation (Swoboda et al., 2022; Basak et al., 2023). They enhance nutrient retention, improve soil mineral composition, increase cation exchange capacity, and contribute to sustainable agricultural fertility management (Manning and Theodoro, 2020; Ramos et al., 2022). As slow-release mineral sources, these powders also help restore nutrient balances, improve water retention, and revitalize soil microbial activity (Anda et al., 2015; Ramos et al., 2022; Guo et al., 2023). This is particularly relevant in semi-arid zones, where conventional fertilizers are often ineffective due to rapid nutrient leaching and unfavourable climatic conditions (Ramos et al., 2022; Cardozo et al., 2024).

**4.5 Effect of trachyte and basalt powder on maize growth and yield**

Trachyte powder, with relatively high concentrations of $K_2O$ and $Na_2O$, promotes plant growth, particularly potassium, which is essential for photosynthesis and water regulation. Its low level of Fe and Mg limits its direct nutrient contribution for maize, which primarily requires nitrogen, phosphorus, and potassium for its growth (Reimann et al., 2018; Linhares et al., 2021). In contrast, basalts, characterized by high concentrations of $Fe_2O_3$, CaO, $TiO_2$, and $P_2O_5$, while exhibiting reduced levels of $SiO_2$ and $Al_2O_3$, can serve as a natural nutrient source (Oppon et al., 2023). They are particularly suited to enriching soils' deficient in these elements, thereby improving soil fertility and productivity (Kagou Dongmo et al., 2018; Luchese et al., 2021; Conceição et al., 2022).

The application of rock powders and conventional fertilizers known as reference fertilization practice in the region (T5) led to an increase in essential parameters related to maize growth and yield, compared to the control (T0). Plots treated solely with trachyte and basalt powders exhibited significantly higher growth compared to control plots, confirming their potential as effective amendments for improving maize growth (Ramos et al., 2015, 2017, 2021, 2022; Oliveira et al., 2020; Mein'da et al., 2022; Luchese et al., 2023; Reis et al., 2024). This improvement could influence photosynthesis, biomass production, and even seed quality, contributing to increased productivity and yield (Oliveira et al., 2020; Viana et al., 2021; Bamberg et al., 2022; Swoboda et al., 2022; Luchese et al., 2023).

There is an overall significantly improvement of maize yield, increasing from 645 kg ha$^{-1}$ in the control to 2763 kg ha$^{-1}$ in the plot treated with trachyte powder and 2931 kg ha$^{-1}$ in the plot amended with basalt powder. Although the highest yield was obtained with the application of conventional fertilizers known as reference fertilization practice in the region recommended by CIFOR-ICRAF (2024) (3164 kg ha$^{-1}$), trachyte and basalt powders could be more advantageous due to their local availability, lower cost, and positive effects on the soil (Tamfuh et al., 2020; Ramos et al., 2020; Mbissik et al., 2023). They



release nutrients slowly and sustainably over multiple seasons, unlike conventional fertilizers, which provide nutrients rapidly but for a short duration. They help improve soil fertility without causing pollution or harming biodiversity (Nkouathio et al., 2008; Lopes et al., 2014; Oliveira et al., 2020; Luchese et al., 2023).

Although no comparison was made between NPK fertilisers and rock powder, rock powder naturally contains a variety of micronutrients essential for plant health, which are often absent in NPK fertilisers. Deficiencies of micronutrients are major

causes of low maize yields in poorly responsive soils (Njoroge et al., 2018). This phenomenon minimizes the agronomic efficiency of N, P and K fertilizers and consequently result in a dwindling economic benefit associated with their use (Njoroge et al., 2018). It is recognised that basalt powder, for example, improve soil fertility and maize nutrition and can help combat plant diseases by strengthening their natural resistance (Conceição et al., 2022). Rock powder can be used in biological agriculture unlike synthetic NPK fertilisers. Rock powder therefore offers a more sustainable and environmentally friendly

approach to soil fertilisation, improving long-term fertility and reducing the risk of pollution (Nkouathio et al., 2008). It represents an attractive alternative to NPK fertilisers for those seeking to adopt more environmentally friendly and sustainable agricultural practices.

## 5 Conclusion

This study aims to evaluate the effect of using trachyte and basalt powders on maize growth and yield in the Guiring area. Geochemically, the Zamai trachytes are rich in silica and potassium, while the basalts are distinguished by their high content of iron, magnesium, and calcium. The soils in Guiring, dominated by a high sand content (62-82%) and low clay (13-23%) and silt (5-15%) contents. Mineralogically, these soils are composed of kaolinite, smectites, sepiolite, and quartz. The pH ranges from 6.8 to 7.2, organic matter (1.3-3.2%), and total nitrogen (0.1%) are low, with a low C/N ratio. The cation exchange

capacity (18.7-25.0 $cmol_c$ $kg^{-1}$ is high, while exchangeable bases ($Ca^{2+}$, $Mg^{2+}$, $K^+$, $Na^+$) and phosphorus are low to moderate, with base saturation varying between 23.6 and 42.4%. The study area has very favourable climatic conditions for maize growth. Soil texture, base saturation, and organic carbon content are classified as S2, indicating moderate limitations. Soil depth (S1-0) is highly suitable, and coarse fragments (S1-1) are slightly suitable. Cation exchange capacity and pH also show slight limitations (S1-1). Overall, the land suitability index is 62.4, classifying it as S2sf, indicating moderate suitability for maize

cultivation. Growth and yield parameters show significant improvement based on the treatments applied. The control treatment ($T_0$) shows the lowest yield, with 645.8 kg $ha^{-1}$. Treatments $T_1$ (2362.9 kg $ha^{-1}$) and $T_2$ (2763.9 kg $ha^{-1}$) showed notable improvements with trachyte powder application. Treatments $T_3$ (2558.6 kg $ha^{-1}$) and $T_4$ (2931.2 kg $ha^{-1}$) highlight the positive effect of basalt powder. Treatment $T_5$ achieves the maximum yield of 3164.5 kg $ha^{-1}$, surpassing all other treatments highlighting an importance of associating urea to rocks powders. Although synthetic fertilizers provide maximum yields in the

short term, trachyte and basalt powders offer a sustainable and cost-effective alternative, improving soils while ensuring competitive yields. Their use is particularly beneficial for resilient and environmentally friendly agriculture. This study will

contribute to the development of regional solutions for soil fertility management, contrasting with the dependency on synthetic fertilizers produced in the Northern Hemisphere, to which tropical countries are reliant. In addition to urea, biological nitrogen fixation processes and phosphorous level may play a crucial role in the agronomic response. This role will be elucidated in the 575 future studies

**Author contributions**

All the authors substantially contributed to this article. The conceptualization of the study was done by DT and BS, with input from CV, SDB. The data acquisition, investigation, methodology, and visualization for the paper were performed by DT, BS, 580 CV, SDB, with substantial input from ANN and MGD. BS and CV wrote the initial draft, and DT and SDB were involved in the reviewing, editing, and the validation of the paper. All authors read and agreed to the final version of the manuscript.

**Data availability**

The research data will be freely available on request from the corresponding author.

**Declarations Competing interests**

On behalf of all authors, the corresponding author states that there is no conflict of interest.

**Acknowledgements**

Authors thank the administrative authorities of the Institute of Agricultural Research for Development (IRAD) of Maroua for providing plots for experimentation and support for field campaigns.

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
