# Peer review of "Soil characteristics, land suitability and effect of trachyte and basalt rock powders on maize (*Zea mays* L.) growth and yield on Fluvisols in Cameroon's Sudano-Sahelian zone (Central Africa)"

_EGUsphere, 2025_

## Referee Comment (RC1)

| Line | Comment |
|---|---|
| L58 | essential nutrients = redundant (plant nutrient = essential element) |
| L58 | enhance its structure: neither of the cited references evaluated soil structure |
| L96 | essential nutrients |
| 2.6 Pedoclimatic Assessment | Not sure what is the purpose of this evaluation |
| Table 4 | There is a surplus line with Mg content after the total major element sum. Maybe the trace elements could be informed in a supplementary material, except maybe for those that are also micronutrients (Zn, Ni and Co). |
| L274-275 | Actually on Table 4 |
| L344 | parent material instead of parent rock |
| L379-380 | Not necessary, since this information was already provided in the methodology. |
| L394 | Actually on Table 10 |
| L430 Table 10 | Please review how the standard deviation for yield was calculated, because the values for all treatments are the same, which is unlikely. |
| L524 | basalts in larger font size |
| Conclusion | Should provide short sentences that summarize the most important results, such as:

The soils in Guiring are dominated by a high sand content (62-82%) and low clay (13-23%) and silt (5-15%) contents and mineralogically, these soils are composed of kaolinite, smectites, sepiolite, and quartz.

The cation exchange capacity (18.7-25.0 cmolc kg-1 is high, while exchangeable bases ($Ca^{2+}$, $Mg^{2+}$, $K^+$, $Na^+$) and phosphorus are low to moderate, with base saturation varying between 23.6 and 42.4%.

The study area has very favourable climatic conditions for maize growth (suitability index of 91.4), while the land suitability index is 62.4, classifying it as S2sf, indicating moderate suitability for maize cultivation. |

| | The control treatment (T0) showed the lowest yield, with 645.8 kg ha-1. Treatments T1 (2362.9 kg ha-1) and T2 (2763.9 kg ha-1) showed notable improvements with trachyte powder application. Treatments T3 (2558.6 kg ha-1) and T4 (2931.2 kg ha-1) highlighted the positive effect of basalt powder. Treatment T5 achieved the maximum yield of 3164.5 kg ha-1. |
| --- | --- |
| L850 | Reference without publication year |

---

## Author Comment (AC1)

**Response to Reviewer Eder Martins**

Following the review, the manuscript showed considerable improvement; however, a few minor adjustments to phrasing and conceptual clarity are still required.

Thank very much for your remarks and suggestions which substantially improve the quality of our manuscript.

The text uses both 'fertiliser' and 'fertilizer'. Please choose one.

Answer: We choose "fertilizer". We also choose "fertilization"

The names of chemical elements must be in lowercase letters.

Answer: Modifications were made in subsection "2.3 Rock sampling and analysis"

Table 4 - Change "Basalte" to "Basalt"

**Answer:** Change was made

**Suggestions**

Lines 29 and 30: Change the original sentence "Although the treatment with conventional fertiliser resulted in a relative higher yield, the advantage of using rock powders lies in their environmental benefits, long-term effectiveness, and more affordable cost" for improved version "Although treatments with conventional fertilizers resulted in comparatively higher yields, rock powders offer significant advantages by facilitating region-specific strategies that deliver environmental benefits, durable effectiveness, and improved affordability.

**Answer: Suggestion adopted**

Lines 66 and 67: Change the original sentence "In Cameroon, in particular, several studies have demonstrated the effectiveness of these powders" for improved version "Multiple studies conducted in Cameroon have demonstrated the effectiveness of these rock powders"

**Answer: Suggestion adopted**

Lines 79 and 80: Change the original sentence "As the minerals constituting rocks dissolve, the released chemical elements become available to plants" for improved version "The chemical elements released during the weathering of rock minerals become available to plants"

**Answer: Suggestion adopted**

Line 481: It is better to use "bioweathering" instead of "biogeochemical".

**Answer:** Suggestion adopted

Lines 521 and 522: Change the original sentence "Trachyte powder, with relatively high concentrations of K2O and Na2O, promotes plant growth, particularly potassium, which is essential for photosynthesis and water regulation." for improved version "Trachyte powder, with relatively high concentrations of K2O and Na2O, promotes plant growth, particularly potassium, which is essential for photosynthesis and water regulation."

**Answer: Suggestion adopted**

Lines 557 and 558: Change the original sentence "The soils in Guiring, dominated by a high sand content (62-82%) and low clay (13-23%) and silt (5-15%) contents." for improved version "The soils in Guiring are dominated by a high sand content (62–82%) and have low clay (13–23%) and silt (5–15%) contents."

**Answer**: Suggestion adopted

---

## Author Comment (AC2)

**Response to Diego Tassinari**

RC1: Comment on egusphere-2025-3474

**General comments**

The manuscript presents a multidisciplinary study that seems to fall well within the scope of the journal, covering fields such as petrography, land evaluation, pedology, soil mineralogy, fertility and a field trial with soil amendments (rock powders or remineralizers). In addition to the very comprehensive characterization of the rocks and the soil, the field trial presents a practical and direct application of this initial assessment, highlighting the specific conditions under which the crops positive response was observed. Although simplistic, with modest sources and doses of fertilizers and remineralizers, the field trial must be evaluated also considering the low availability of published data from the studied region, the amount of work needed to grind dozens of kilograms of rock and the significant responses obtained. In addition, the practical applications of the results for this region and elsewhere are very significant, as they deal with fertilizer shortage and food security. What may seem as major setbacks of the study are the lack of plant nutrient contents to show how nutrient uptake responded to the treatments and the field trial restricted to a single crop cycle. Regarding the latter, it must be pointed out that significant differences in yield were already perceived in this first cycle.

Answer: We sincerely thank you, Sir, for this very pertinent remark. We acknowledge that our study did not consider the nutrient content of maize plants, nor did it directly evaluate foliar or grain composition, which limits the precise assessment of plant responses to the different treatments in terms of nutrient uptake. Nevertheless, the agronomic indicators used (yield and growth parameters) provide relevant insights, and the integration of nutritional analyses will be an important perspective for future research.

**Specific comments**

Treatment application is not clear enough. It is important to understand how the rock powder was applied, with broadcasted in the entire plot or locally applied and whether it was incorporated or not by any tillage practice.

Answer: Dear Sir, we sincerely thank you for your very pertinent question regarding the application of rock powder treatments. We would like to clarify that, after the establishment of the experimental setup and seedling emergence, the rock powder was applied in a localized manner, directly into each planting hole. It was then incorporated at a depth of approximately 5 cm, in order to minimize leaching by water and dispersion by wind. This localized incorporation enhanced the availability of nutrients at the root level while reducing losses. This clarification has been added to the manuscript in line 139-142, in the subsection « 2.2 Experimental design, treatments, and plant material ».

Yield results could be presented also as relative yield, especially in the discussion, conclusion and abstract, because it may be more directly referred by other studies. For example, for NPK + urea as 100% relative yield, basalt + urea reached 92.6% and trachyte + urea reached 87.3% of the maximum yield, whereas the remineralizers alone resulted in relative yields of 80.8% and 74.7% for basalt and trachyte respectively.

Answer: Suggestion adopted. This information was added in lines 540-544 in the discussion section, in the conclusion in lines 571-575 and in the abstract in lines 27-30.

**Technical corrections**

| Line | Comment                                                                                                                                                                                                                                                                                                                                            |
|------|----------------------------------------------------------------------------------------------------------------------------------------------------------------------------------------------------------------------------------------------------------------------------------------------------------------------------------------------------|
| L58  | essential nutrients = redundant (plant nutrient = essential element)  Answer: "essential nutrients" was replaced by "plant nutrients."                                                                                                                                                                                                             |
| L58  | enhance its structure: neither of the cited references evaluated soil structure  Answer: The following reference was added: Buss, W., Hasemer, H., Ferguson, S. and Borevitz, J. Stabilisation of soil organic matter with rock dust partially counteracted by plants. Global Change Biology, 30, e17052. https://doi.org/10.1111/gcb.17052, 2024. |
|      | essential nutrients                                                                                                                                                                                                                                                                                                                                |

| L96                          | Answer: "essential nutrients" was replaced by "essential elements."                                                                                                                                                                                                                                                                                                                                                                                                                                                                                                                                                                                                                                                                                                                                                                                                      |
|------------------------------|---------------------------------------------------------------------------------------------------------------------------------------------------------------------------------------------------------------------------------------------------------------------------------------------------------------------------------------------------------------------------------------------------------------------------------------------------------------------------------------------------------------------------------------------------------------------------------------------------------------------------------------------------------------------------------------------------------------------------------------------------------------------------------------------------------------------------------------------------------------------------------|
| 2.6 Pedoclimatic
Assement | Answer: Dear Sir, thank you very much. Modifications were made in red in lines 222-225 as follows:  The pedoclimatic assessment aimed to determine the suitability of the study area for maize cultivation based on climatic data (precipitation, temperature, relative humidity, and insolation) collected at the Maroua-Salak station (Cameroon, 10°27'0" north latitude and 14°15'0" east longitude) between 1980 and 2020, and the pedological characteristics of soils such as topography, flooding, texture, depth, and cation exchange capacity, base saturation, organic carbon, pH and slinity, in accordance with the climatic and requirements of crops (Sys, 1985; Sys et al., 1993; Issiné et al 2022). It allows to identify the potential limiting factors for maize cultivation. A climatic index (CI) was calculated using the parametric formula (Sys, 1985): |
| Table 4                      | There is a surplus line with Mg content after the total major element sum. Maybe the trace elements could be informed in a supplementary material, except maybe for those that are also micronutrients (Zn, Ni and Co).  Answer: Dear Sir, thank you very much. This line was deleted. It was Mg# used for magmatic differentiation.                                                                                                                                                                                                                                                                                                                                                                                                                                                                                                                                            |
| L274-275                     | Acrually on Table 4 Answer: "Table 3" was replaced by "Table 4"                                                                                                                                                                                                                                                                                                                                                                                                                                                                                                                                                                                                                                                                                                                                                                                                          |
| L344                         | Parent material instead of parent rock Answer: "parent rock" was replaced by "parent material."                                                                                                                                                                                                                                                                                                                                                                                                                                                                                                                                                                                                                                                                                                                                                                          |
| L379-380                     | Not necessary, since this information was already provided in the methodology.  Answer: Dear Sir, this information was deleted                                                                                                                                                                                                                                                                                                                                                                                                                                                                                                                                                                                                                                                                                                                                                  |
| L394                         | Actually Table 10  Answer: "Table 9" was replaced by "Table 10"                                                                                                                                                                                                                                                                                                                                                                                                                                                                                                                                                                                                                                                                                                                                                                                                                 |

| L430
Tableau 10 | Please review how the standard deviation for yield was calculated, because the values for all treatments are the same, which is unlikely.  Answer: We sincerely thank you Sir. There has been a reporting error. Modifications were made as follows: $T_0=645.8\pm65.0a$ ; $T_1=2362.9\pm120.0b$ ; $T_2=2763.9\pm140.0c$ ; $T_3=2558.6\pm130.0d$ ; $T_4=2931.2\pm150.0^e$ ; $T_5=3164.5\pm160.0f$ |
|--------------------|---------------------------------------------------------------------------------------------------------------------------------------------------------------------------------------------------------------------------------------------------------------------------------------------------------------------------------------------------------------------------------------------------|
|                    | basalts in larger font size                                                                                                                                                                                                                                                                                                                                                                       |
| L524               | Answer: Correction was made.                                                                                                                                                                                                                                                                                                                                                                      |
|                    | Should provide short sentences that summarize the most important results, such as:                                                                                                                                                                                                                                                                                                                |
|                    | The soils in Guiring are dominated by a high sand content (62-82%) and                                                                                                                                                                                                                                                                                                                            |
|                    | low clay (13-23%) and silt (5-15%) contents and mineralogically, these                                                                                                                                                                                                                                                                                                                            |
|                    | soils are composed of kaolinite, smectites, sepiolite, and quartz.                                                                                                                                                                                                                                                                                                                                |
|                    | The cation exchange capacity (18.7-25.0 cmolc kg-1 is high, while                                                                                                                                                                                                                                                                                                                                 |
|                    | exchangeable bases (Ca 2+ , Mg 2+ , K + , Na + ) and phosphorus are low to                                                                                                                                                                                                                                                                            |
|                    | moderate, with base saturation varying between 23.6 and 42.4%.                                                                                                                                                                                                                                                                                                                                    |
| Conclusion         | The study area has very favourable climatic conditions for maize growth (suitability index of 91.4), while the land suitability index is 62.4,                                                                                                                                                                                                                                                    |
|                    | classifying it as S2sf, indicating moderate suitability for maize                                                                                                                                                                                                                                                                                                                                 |
|                    | cultivation.                                                                                                                                                                                                                                                                                                                                                                                      |
|                    | The control treatment (T0) showed the lowest yield, with 645.8 kg ha-                                                                                                                                                                                                                                                                                                                             |
|                    | 1. Treatments T1 (2362.9 kg ha-1) and T2 (2763.9 kg ha-1) showed                                                                                                                                                                                                                                                                                                                                  |
|                    | notable improvements with trachyte powder application. Treatments T3                                                                                                                                                                                                                                                                                                                              |
|                    | (2558.6 kg ha-1) and T4 (2931.2 kg ha-1) highlighted the positive effect                                                                                                                                                                                                                                                                                                                          |
|                    | of basalt powder. Treatment T5 achieved the maximum yield of 3164.5                                                                                                                                                                                                                                                                                                                               |
|                    | kg ha-1.                                                                                                                                                                                                                                                                                                                                                                                          |

|      | Answer: Thank you very much. All your suggestions have been |
|------|--------------------------------------------------------------------|
|      | taken into consideration. We also add relative yields as suggested |
|      | above.                                                             |
|      | Reference without publication year                                 |
|      | Answer: The publication year "2024b" was added.                    |
| L850 |                                                                    |

---

## Author Comment (AC3)

**Reponse to reviewer RC2**

**Major Revisions**

**1- Title and Abstract**

The title and abstract should be revised to accurately reflect the main findings of the study. As currently written, they partially present the experimental results, lacking emphasis on pedological information and crop suitability.

**Title**

Answer: Thank you very much for your suggestion. The title has been revised as follows: "Soil characteristics, land suitability and effect of trachyte and basalt rock powders on maize (Zea mays L.) growth and yield on Fluvisols in Cameroon's Sudano-Sahelian zone (Central Africa)".

**Abstract**

Answer: The sentence "These soil characteristics are moderately suitable for maize cultivation" was replaced with the following sentence "The study area has very favourable climatic conditions for maize growth (suitability index of 91.4), with a land suitability index of 62.4, classifying it as S2sf, indicating moderate suitability for maize cultivation "in lines 25-26.

**2- Introduction**

The introduction should be restructured to better reflect the reality of soil fertility management in Africa, where fertilizer use remains low and nutrient depletion is the dominant challenge. To structure the introduction around "intensive agriculture" does not align with the general conditions of sub-Saharan smallholder systems.

Answer: Thank you very much for your comment. We have taken your recommendation into account and restructured the introduction as follows at the beginning of the introduction in lines 16–50 " In Sub-Saharan Africa, soil degradation represents a major obstacle to food security and the sustainability of agricultural production systems. Increasing population pressure on arable land, combined with extensive farming practices and low input, leads to the gradual depletion of essential soil nutrients (Jhariya et al., 2021; Khatri et al., 2024). In most rural areas, agriculture remains predominantly subsistence-based, relying on traditional techniques and limited mineral inputs due to the high cost and restricted accessibility of synthetic fertilizers (Sinha et al., 2022). These conditions result in declining yields, deteriorating soil structure, and loss of organic matter (Nanganoa et al., 2019; Rajwar et al., 2021).

In recent decades, population growth and food insecurity have led to an intensification of extensive farming practices and increased of soil overexploitation (Pham et al., 2018; Jhariya et al., 2021; Khatri et al., 2024). This situation is further exacerbated in addition to limited access to inputs, by logistical constraints and institutional challenges, causing a continuous decline in soil fertility and crop productivity (Kuria et al., 2018; Nanganoa et al., 2019). As a limited resource, soil takes between 200 to 1000 years to form a layer of 2.5 cm in thickness, making its exploitation for agricultural purposes fragile due to the increase in the world population and climate change (Moges and Taye, 2016). Soil degradation thus represents a major environmental and socio-economic challenge, impacting biodiversity, ecosystem services, and food security. Sustainable management requires integrated strategies aimed at restoring or maintaining nutrient balance and enhancing agroecosystem resilience (Nkouathio et al., 2008; Mekuriaw et al., 2017; Hossain et al., 2020; Garai et al., 2021)."

**3- Integration of Pedological Information / crop suitability**

The manuscript includes a detailed pedological study and soil and crop suitability assessment, but this information was not integrated into the experimental design or discussion (example: treatment based on soil organic amendment to improve the water retention in sandy soils). The authors should link the soil information gener8ated in the first section to treatment responses, explaining how these factors may have influenced maize growth and yield.

Answer: Dear Sir, thank you very much for your comment. The soil information contained in the first section is indeed linked to the treatment responses. It had influenced the growth and yield of maize in the control plot. This control plot had the lowest growth parameters and yields primarily due to constraints related to the soil's physical properties (texture) and fertility (base saturation and organic carbon) as the soil was classified as S2sf, indicating moderate suitability for maize cultivation. It was demonstrated that clay fraction (13 to 18%) and the presence of smectite improve for example the water retention in the study sandy soil in the first section of the discussion. Also, in the introduction, it is clearly stated that numerous studies have explored the use of silicate rock powders for soil fertilization and remineralization. In the Americas for example, several studies have demonstrated that these mineral powders enhance water retention and soil fertility while contributing to carbon sequestration (Beerling et al., 2018; Theodoro et al., 2021; Lewis et al., 2021; Ramos et al., 2022; Medeiros et al., 2023).

**4- Section on Nutrient Release Kinetics**

The section discussing the kinetics of nutrient release from trachyte and basalt powders lacks supporting experimental data. Since no direct measurements of nutrient release dynamics or plant uptake were conducted, this section should be removed or significantly shortened to avoid overinterpretation.

Answer: Thank you very much for the suggestion. Please Sir, this section was recommended by some reviewer in the first round of interactive discussion. Nevertheless, the following three sentences (59 words) which are not directly linked to silicate dissolution and nutrient release were deleted:

- Meena and Biswas (2014) demonstrated that rock phosphate and mica release cations more rapidly when finely ground and combined with compost.
- Bacillus mucilaginosus produces organic acids that enhance nutrient solubilization (Yang et al., 2016).
- Root exudates and mycorrhizal associations further improve K and Mg bioavailability, facilitating plant uptake (Ma and Yamaji, 2006; Menna and Biswas, 2014).

**Specific Comments**

5- Lines 20–25: Improve this section by summarizing the dominant soil clay minerals and overall texture and remove overly descriptive details that are not directly relevant to the main results.

Answer: Dear Sir, thank you for your suggestion. The section has been reformulated as follows in lines 21-24: " The soil is sand-dominated with a neutral pH, and its clay fraction is composed of kaolinite, smectites, and sepiolite. It exhibits low levels of organic matter (2.6%) and nitrogen (0.1%) but has a moderate-to-high cation exchange capacity (22.1 cmolc kg-1) and an available phosphorus content (19 mg kg-1). This soil is classified as Ochric Dystric Fluvisols according to the WRB ".

6- Lines 35–40: The general statement does not accurately reflect the current reality of African agriculture. Please rephrase to reflect context-specific challenges, such as low fertilizer use, limited access to inputs, political issues, logistics constraints, and soil nutrient depletion.

Answer: Thank you, Sir, for this remark. Revision was done also at the following of your above recommendation concerning the introduction as follows:

"In recent decades, population growth and food insecurity have led to an intensification of extensive farming practices and increased of soil overexploitation (Pham et al., 2018; Jhariya et al., 2021; Khatri et al., 2024). This situation is further exacerbated in addition to limited access to inputs, by logistical constraints and institutional challenges, causing a continuous decline in soil fertility and crop productivity (Kuria et al., 2018; Nanganoa et al., 2019). As a limited resource, soil takes between 200 to 1000 years to form a layer of 2.5 cm in thickness, making its exploitation for agricultural purposes fragile due to the increase in the world population and climate change (Moges and Taye, 2016). Soil degradation thus represents a major environmental and socio-economic challenge, impacting biodiversity, ecosystem services, and food security. Sustainable management requires integrated strategies aimed at restoring or maintaining nutrient balance and enhancing agroecosystem resilience (Nkouathio et al., 2008; Mekuriaw et al., 2017; Hossain et al., 2020; Garai et al., 2021)."

**7- Lines 45–50: Clarify the conceptual difference between *soil health* and *soil quality*.**

Answer: Thank you very much for this important remark. Dear Sir, soil quality refers to the inherent, measurable physical and chemical properties of a soil, whereas soil health is the dynamic, holistic capacity of a soil to function as a vital living ecosystem. Soil quality is a component of soil health. The following sentence (lines 45-47) " Soil health, also known as soil quality, is a key factor in sustainable agriculture, influencing ecosystem quality, such as air and water quality" was revised as follows:

" Soil health is a key factor in sustainable agriculture, influencing ecosystem quality, such as air and water quality"

8- Lines 50–55: The statement suggesting that intensive agriculture is a major problem in Africa is misleading. In most African systems, the challenge is low agricultural intensification rather than overuse of inputs.

Answer: Thank you very much. Sir, the expression "intensive agriculture" was replaced by "Intensification of extensive farming practices"

9- Line 121: Remove the duplicated reference to "(Table 1)".

Answer: The duplicated reference to "(Table 1)" was removed

10- Line 135: Avoid using abbreviations such as "IRAD" at their first occurrence; write the full name before introducing the acronym.

Answer: Thank you for this remark. The first occurrence of the acronym "IRAD" appears at line 112, where it is properly defined as: "Agricultural Research Institute for Development (IRAD)"

11- Line 290: Add in Table 4, the international regulatory limits for trace elements in soil amendments and fertilizers, for proper environmental assessment.

Answer: Thank you for this suggestion. We add standard limits according to FAO/WHO for the following trace elements in agricultural soils: V (50 mg/kg), Cr (100 mg/kg), Co (20 mg/kg), Ni (50 mg/kg), Zn (200 mg/kg), Pb (50 mg/kg), U (20 mg/kg). We also add Standard limits according to Environment protection agency United State (EPA) for major elements in agricultural soils: Si (40%), Al (8%), Fe (5%), Mn (0.3%), Mg (1.5%), Ca (3%), Na (1.5%), K (3%), P (0.2%).

12- Line 327: In Table 6, avoid introducing abbreviations such as "TEB" before defining them.

Answer: Thank you, Sir, for this observation. The term "TEB" was defined at its first occurrence in line 195, as "Total exchangeable bases" before being used again in the table 6.

13- Line 480: The section titled "4.3 Kinetics of nutrient element release from trachyte and basalt powders" is overstated. The experimental design does not allow evaluation of release kinetics, plant uptake, or nutrient dynamics. This section should be removed.

Answer: Thank you very much for your suggestion. As already stated above in the Major revision section, this section was recommended by some reviewer in the first round of interactive discussion. Nevertheless, the following three sentences (59 words) which are not directly linked to silicate dissolution and nutrient release were deleted:

- Meena and Biswas (2014) demonstrated that rock phosphate and mica release cations more rapidly when finely ground and combined with compost;

- Bacillus mucilaginosus produces organic acids that enhance nutrient solubilization (Yang et al., 2016).
- Root exudates and mycorrhizal associations further improve K and Mg bioavailability, facilitating plant uptake (Ma and Yamaji, 2006; Menna and Biswas, 2014).

14- Line 508: The suggestion to use amendments based on clay materials is not cost-effective for smallholder farmers. Given the applied nature of the research, the socio-economic feasibility of proposed solutions should be considered in the recommendations.

Answer: Dear Sir, thank you for this comment. We deleted the suggestion to use amendments based on clay materials. The sentence was revised with an addition of one reference as follows: "Adding biochar with and without additional compost and manure, identified by Hussain et al. (2021) and Seyedsadr et al. (2022) as an effective solution, enhances water retention and cation exchange capacity, both of which are fundamental properties for the fertility of sandy soils "

---

## Author Comment (AC7)

**Response to Reviewer RC3 \_ Eder Martins**

After revising the text, the article is ready for publication.

Answer: Dear reviewer, thank you very much for your comments and suggestions which substantially improved the quality of our paper.

---

## Author Response (AR1)

**Response to Editor**

1- L39: Please cite direct sources for the soil formation rates.

> **Answer:** **The two following references "Stockmann et al. (2014)" and "Zhang et al. (2024)" were added in line 45.**

> **Stockmann, U., Minasny, B. and McBratney, A. B.: How fast does soil grow? Geoderma 216, 48–61. http://dx.doi.org/10.1016/j.geoderma.2013.10.007, 2014.**

> **Zhang, Y., Hartemink, A. E., Vanwalleghem, T., Bonfatti, B. R. and Moen, S.: Climate and land use changes explain variation in the A horizon and soil thickness in the United States. Communications Earth & Environment 5, 129. https://doi.org/10.1038/s43247-024-01299-6, 2024.**

2- Table 10: In your response to referee #1, the corrected standard deviation values for maize yield are rounded, while the treatment means are not. Please standardise—I suggest removing the decimal case.

> **Answer:** **Suggestion adopted. Corrections were made in the table 10.**

3- Please also correct the maize yield standard deviation values in the text (lines 422-429).

> **Answer:** **Corrections made in lines 435-439 and even in other parts of the manuscript.**

---

## Author Response (AR2)

**Reponse to Editor**

We thank you, Sir, for your recommendation. The suggested technical corrections have been made, and we are grateful for your valuable feedback. Also, copy-editing was done and all modifications are in red in the manuscript.

**Topic editor : Pedro Batista**

**L15:** At the Guiring experimental farm [add the location, for example in the Far North Region of Cameroon or in the Sudano-Sahelian zone of Cameroon].
**Answer :** **The location "in the Far North Region of Cameroon" was added in line 15.**

**L18:** List the treatments.
**Answer:** **Dear Sir, treatments were added in lines 19 - 24.**

**Abstract:** Delete the sentence: *"The study area presents very favorable climatic conditions for maize cultivation (suitability index of 91.4), with a soil suitability index of 62.4, classifying it as S2sf, which indicates moderate suitability for maize cultivation."*
**Answer:** **The sentence *was deleted***

**L26–30:** Remove decimal values.
**Answer:** **The decimal values were removed in lines 30-32.**

**L37:** Delete "essential."
**Answer:** **The word *"essential"* was deleted in line 40**.

**L120:** Add the objectives at the end of the introduction.
**Answer:** **The objectives of the study were added in lines 124-129**

**Map:** Add altitude values to the contour lines; otherwise, they are not informative. I also suggest removing the roads.

**Answer: The altitude values of the contour lines and spot heights were added. Roads were also removed from the map.**

**L142:** Delete the sentence *"It was conducted at the IRAD experimental site in Guiring."* This has already been mentioned.

**Answer:** *The sentence was deleted in line 151.*

**L337:** Revise/correct the sentence: *"No correlation was observed between pH and the other elements, except for clay (r = 0.9, p < 0.05)."*

**Answer:** **The sentence was revised as follows:** *"Except for clay, which was strongly positively correlated with pH (r = 0.9, p < 0.05), no significant correlations were observed between pH and other soil parameters"* **in lines 347-349.**

**L378:** The suitability classes (S1-1, FO) are mentioned in the text without being explained. In Tables 8 and 9, add a legend for the acronyms and suitability classes.

**Answer:** **Thank you very much for the suggestion. A legend for the acronyms and suitability classes was added in Tables 8 and 9.**

**Table 8:** Remove decimal values from the precipitation values (total?) (mm). Add n/N to the legend.

**Answer:** **The decimal values were removed from the precipitation values and n/N was added to the legend in Table 8.**

**L425:** Where is the time-based response indicated in Table 10?

**Answer:** **The expression "over time" was deleted in line 438**

**L517:** Revise the sentence: *"Improving soil texture is also an essential strategy."* This sentence is not relevant here.

**Answer:** **Thank you very much for this suggestion. The sentence** *"Improving soil texture is also an essential strategy"* **was deleted in line 529.**

**L568:** In the Guiring area, [add the location as above].

**Answer : The location "in the Far North Region of Cameroon" was added in line 581.**

**Executive editor : Rémi Cardinael**

I also think the manuscript would benefit a lot from some copy-editing, and kindly ask you to do so.

Answer: We went through the paper for copy-editing. All modifications are in red in the manuscript. They are presented in the following lines.

Line 45. " and increase of " was replaced by " and an increase in "

Line 56. " quality " was replaced by " qualities "

Line 64. " Intensification " was replaced by " Also, the intensification "

Line 95. " limiting " was replaced by " thereby limiting "

Line 247. " slinity " was replaced by " salinity "

Line 248. " climatic " was replaced by " climatic conditions "

Line 265. " from each plot an average value " was replaced by " from each plot and an average value "

Line 291. " by elongated, euhedral prisms " was replaced by " by elongated euhedral prisms "

Line 300. " The $SiO_2$ content ranges between 65.3% and 66.5%. Meanwhile the $TiO_2$ content varies from 0.2% to 0.3% " was replaced by " The $SiO_2$ content ranges between 65.3% and 66.5%, meanwhile the $TiO_2$ content varies from 0.2% to 0.3% "

Line 333. " reveal " was replaced by " revealed "

Line 369. " different soil texture " was replaced by " different soil textures "

Line 371. " texture " was replaced by " textures "

Line 373. " colour of soil " was replaced by " colour of soils "

Line 407. " good uniformity and a of plant growth " was replaced by " good uniformity of plant growth "

Line 454. " increase " was replaced by " increases "

Line 557. " significantly " was replaced by " significant "

Line 571. " result " was replaced by " results "

Line 572. " improve " was replaced by " improves "

Line 584. " acid and neutral with values " was replaced by " acid and neutral, with values "

Line 600. " in the future studies " was replaced by " in future studies. "